# Reduction of endocytosis and EGFR signaling is associated with the switch from isolated to clustered apoptosis during epithelial tissue remodeling in *Drosophila*

Kevin Yuswan[1], Xiaofei Sun[1], Erina Kuranaga[1,2]*, Daiki Umetsu[3]*

**1** Laboratory for Histogenetic Dynamics, Graduate School of Life Sciences, Tohoku University, Sendai, Japan, **2** Laboratory for Histogenetic Dynamics, Graduate School and Faculty of Pharmaceutical Sciences, Kyoto University, Kyoto, Japan, **3** Laboratory of Cell Biology, Department of Biological Sciences, Graduate School of Science, Osaka University, Osaka, Japan

* erina.kuranaga.d1@tohoku.ac.jp (EK); daiki.umetsu.sci@osaka-u.ac.jp (DU)

## Abstract

Epithelial tissues undergo cell turnover both during development and for homeostatic maintenance. Removal of cells is coordinated with the increase in number of newly dividing cells to maintain barrier function of the tissue. In *Drosophila* metamorphosis, larval epidermal cells (LECs) are replaced by adult precursor cells called histoblasts. Removal of LECs must counterbalance the exponentially increasing adult histoblasts. Previous work showed that the LEC removal accelerates as endocytic activity decreases throughout all LECs. Here, we show that the acceleration is accompanied by a mode switching from isolated single-cell apoptosis to clustered ones induced by the endocytic activity reduction. We identify the epidermal growth factor receptor (EGFR) pathway via extracellular-signal regulated kinase (ERK) activity as the main components downstream of endocytic activity in LECs. The reduced ERK activity, caused by the decrease in endocytic activity, is responsible for the apoptotic mode switching. Initially, ERK is transiently activated in normal LECs surrounding a single apoptotic LEC in a ligand-dependent manner, preventing clustered cell death. Following the reduction of endocytic activity, LEC apoptosis events do not provoke these transient ERK up-regulations, resulting in the acceleration of the cell elimination rate by frequent clustered apoptosis. These findings contrasted with the common perspective that clustered apoptosis is disadvantageous. Instead, switching to clustered apoptosis is required to accommodate the growth of neighboring tissues.

## Introduction

The development of multicellular organisms is a tight, spatiotemporally regulated process that results in the proper shapes and sizes of adult bodies while maintaining the viability of multicellular organisms during development itself. Throughout this process, continuous and systemic changes in internal and external factors challenge the chemical and mechanical balance

**Data Availability Statement:** All numerical data are found in excel files as Supplementary files. In

addition, all data and codes are available in a repository site linked below which includes the DOI number as described in the Materials and Methods. URL - https://github.com/k0yuswantohoku/LEC_removal_simulation. DOI generated via Zenodo (DOI 10.5281/zenodo.13290047) and shown in the release notes and "README - DOI and Repository Description.md" file.

**Funding:** This work was supported by the following grants: Japan Science and Technology Agency - JST SPRING (JPMJSP2114) to K.Y., Core Research for Evolutional Science and Technology - JST CREST (JPMJCR1852), Japan Society for the Promotion of Science - MEXT KAKENHI (JP26114003, JP21H05255), JSPS KAKENHI (JP24687027, JP16H04800), Japan Agency for Medical Research and Development - AMED moonshot (22zf0127001h002) and the research grant for Astellas Foundation for Research on Metabolic Disorders to E. K., JSPS KAKENHI (JP21K06144, JP21H05105, JP24H01405), JST FOREST Program (J210000474), Takeda Science Foundation, and Tohoku University FRIS Program for Creation of Interdisciplinary Research to D. U. The funders had no role in study design, data collection and analysis, decision to publish, or preparation of the manuscript.

**Competing interests:** The authors have declared that no competing interests exist.

**Abbreviations:** AP2, adaptor protein-2; CHC, clathrin heavy chain; dsRNA, double-stranded RNA; EGFR, epidermal growth factor receptor; ERK, extracellular-signal regulated kinase; hAPF, hours after puparium formation; LEC, larval epidermal cell; RTK, receptor tyrosine kinase; ROI, regions of interest; TLR, Toll-like receptor.

of various tissues to promote their remodeling [1–5]. One extreme example of tissue remodeling is the removal of larval cells via apoptosis and the subsequent replacement by adult cells in multiple organs in the metamorphosis of some vertebrates and invertebrates [6–10]. For example, intestinal remodeling in *Xenopus* takes place during the climax of metamorphosis, characterized by the shortening of the intestine within 8 days. It starts off with the elimination and dedifferentiation of the larval intestinal cells, followed by the rapid expansion of the thicker and folded adult intestinal epithelial layer [8,9]. *Drosophila melanogaster* is an excellent model to examine mass cell elimination during development. One of the most easily observable tissue replacement process occurs at the abdominal epithelium, where polyploid larval epidermal cells (LECs) undergo large-scale replacement by adult histoblasts [11]. Histoblasts remain quiescent throughout the embryonic and larval stages at the histoblast nests located in the lateral abdomen [11]. At around 15 hours after puparium formation (hAPF), mass proliferation of the histoblasts commences, leading to the expansion of the histoblast nests. In coordination with the rapidly proliferating histoblasts, the number of apoptotic LECs increases [12,13]. Starting at 20 hAPF, the early phase of LEC apoptosis occurs infrequently for around 5 h [14]. This stage is then followed by the late phase, when LECs migrate toward the dorsal midline and undergo rapid elimination until their complete replacement by histoblasts [14,15]. Overall, this process takes between 15 and 20 h to complete. Importantly, the removal of LECs must counterbalance the exponentially increasing histoblasts by accelerating the cell elimination rate. However, the precise cellular and molecular mechanism by which cell elimination rate accelerates during this process remains elusive.

LEC apoptosis is caspase-dependent and progresses through apical constriction and extrusion, along with the expansion of histoblasts [12–14,16,17]. During this time, the actomyosin cables accumulate at the cell cortex, which constricts the apical surface of the cell [18,19]. Recent work in our laboratory revealed the network of feedback mechanisms whereby the reduction of endocytic activity promotes both the accumulation of cortical actomyosin and the decrease of junctional E-cadherin, leading to a further decrease in endocytic activity [14]. This reduction stimulates caspase activation, resulting in caspase-dependent apoptosis of LECs. As an essential regulator of the signal transduction of various molecular pathways, endocytosis regulates signaling pathways including Toll-like receptors (TLRs), Wnt receptors, as well as receptor tyrosine kinases (RTKs) [20–22]. One such RTK functioning in epithelial cells is the epidermal growth factor receptor (EGFR). EGFR has been well established as a regulator of epithelial cell survival, growth, and proliferation in its activation of various downstream pathways, including the MAPK cascades [23–26]. Studies have demonstrated the activation by phosphorylation of EGFR within clathrin-mediated endosomes [27,28], making EGFR a promising candidate signaling pathway that may also be regulated by endocytic activity in LECs. Recent studies showed that in developing *Drosophila* epithelial tissues, EGFR signaling is down-regulated during cell compaction and extrusion while its up-regulation induces cell survival [29–31]. Elucidating whether EGFR is regulated by endocytosis, and the biological role of spatiotemporal regulation of EGFR in LECs is essential to understand the molecular mechanisms of massive cell elimination during the epithelial tissue remodeling. Here, we discovered that LEC elimination is regulated by molecular and cellular interactions at a local cell cluster level, mediated by the EGFR/ERK signaling pathway. Using an ERK activity reporter, we found that during the early phase ERK activity is transiently up-regulated in cells surrounding a single apoptotic LEC. In contrast, these transient events are diminished in low endocytic activity and EGFR signaling conditions, promoting an increase of clustered LEC eliminations. We found that the ERK up-regulation events are ligand dependent, as the knockdown of EGFR ligand *vein* leads to the loss of transient activation and, consequently, the increase of cluster elimination frequency. Overall, this study portrays cross-scale interactions

between molecular, cellular, and intercellular level regulatory systems that lead to the tight temporal regulation of massive cell eliminations during the tissue remodeling.

## Results

### EGFR signaling pathway limits LEC elimination rate and acts downstream to endocytosis

Our previous study has shown that endocytic activity decreases throughout LEC elimination as a means to regulate the number of cells undergoing caspase activation. Since the down-regulation of EGFR signaling induces caspase activation in *Drosophila* and other systems [26,29,32,33], we considered the EGFR signaling pathway as a strong candidate mechanism to be potentially modulated by endocytic activity during the LEC elimination process. To investigate whether EGFR signaling has a role in maintaining the LEC elimination rate, we either genetically knocked-down or overexpressed EGFR in the majority of LECs under the control of *pnr*-GAL4, except for LECs near the segmental boundary (Fig 1A). Our analysis also did not include the LECs bordering the histoblasts, which are preferentially eliminated around the initiation of histoblast expansion as previously reported [12]. LEC elimination viewed from the dorsal side progresses in 2 phases: the early phase (20 to 25 hAPF) is characterized by infrequent elimination while the late phase (25 to 40 hAPF) with rapid LEC migration toward the midline and massive LEC death (Fig 1A and 1B; [14]). Using these as a reference, we evaluated dorsal LEC elimination in the early and late phases. We knocked down EGFR (*EGFR*$^{\text{RNAi}}$) by expressing double-stranded RNA (dsRNA). *EGFR* $^{\text{RNAi}}$ initiated the massive elimination of LECs already during the early phase (Fig 1C and 1E, red line). Consistent with this result, the expression of a constitutively activated form of EGFR (*EGFR*$^{\lambda\text{top}}$) [34] diminished LEC elimination (Fig 1D and 1E, green line). These results suggest that EGFR inhibits LEC elimination. To examine whether EGFR signaling is acting downstream to endocytic activity, we blocked endocytic activity while constitutively activating EGFR via the expression of a temperature-sensitive, dominant-negative form of *shibire* (*shi*$^{\text{TS}}$) [35] and *EGFR*$^{\lambda\text{top}}$. To accommodate the developmental delay due to temperature shift from 18˚C to 29˚C, we started our imaging from 30 hAPF (Fig 1F, see Materials and methods). While *shi*$^{\text{TS}}$ alone promoted LEC elimination (Fig 1G and 1I, dark blue line), simultaneous expression of *EGFR*$^{\lambda\text{top}}$ with *shi*$^{\text{TS}}$ suppressed LEC elimination (Fig 1H and 1I, light blue line). These results indicate that EGFR activity has an inhibitory effect against LEC elimination and works downstream to endocytic activity.

### EGFR signaling activity decreases over time in LECs in an endocytosis-dependent manner

Our genetic experiments showed that EGFR activity is downstream to endocytosis, suggesting that EGFR undergoes down-regulation along with the reduction of endocytic activity throughout the LEC elimination period. In order to determine whether EGFR signaling activity reduces throughout LEC elimination similar to endocytotic activity, we monitored EGFR signaling activity by tracking an ERK signaling reporter, miniCic, tagged with mScarlet (miniCic::mScarlet) [29]. In high ERK activity, miniCic localizes in the cytoplasm, while low ERK conditions promote the nuclear translocation of miniCic (Figs 2A and S1A–S1D). This sensor revealed that the average ERK activity in LECs at the beginning of the early phase (20 h) begins at a low level, as shown by the nuclear localization of miniCic (Fig 2B, 2C, and 2F). This nuclear intensity further increased over time throughout the early and late phases (Figs 2B, 2C, 2F, and S1E). This observation shows that there is a global ERK activity decrease throughout the LEC elimination period. The dynamics of the decrease in ERK activity parallels with the

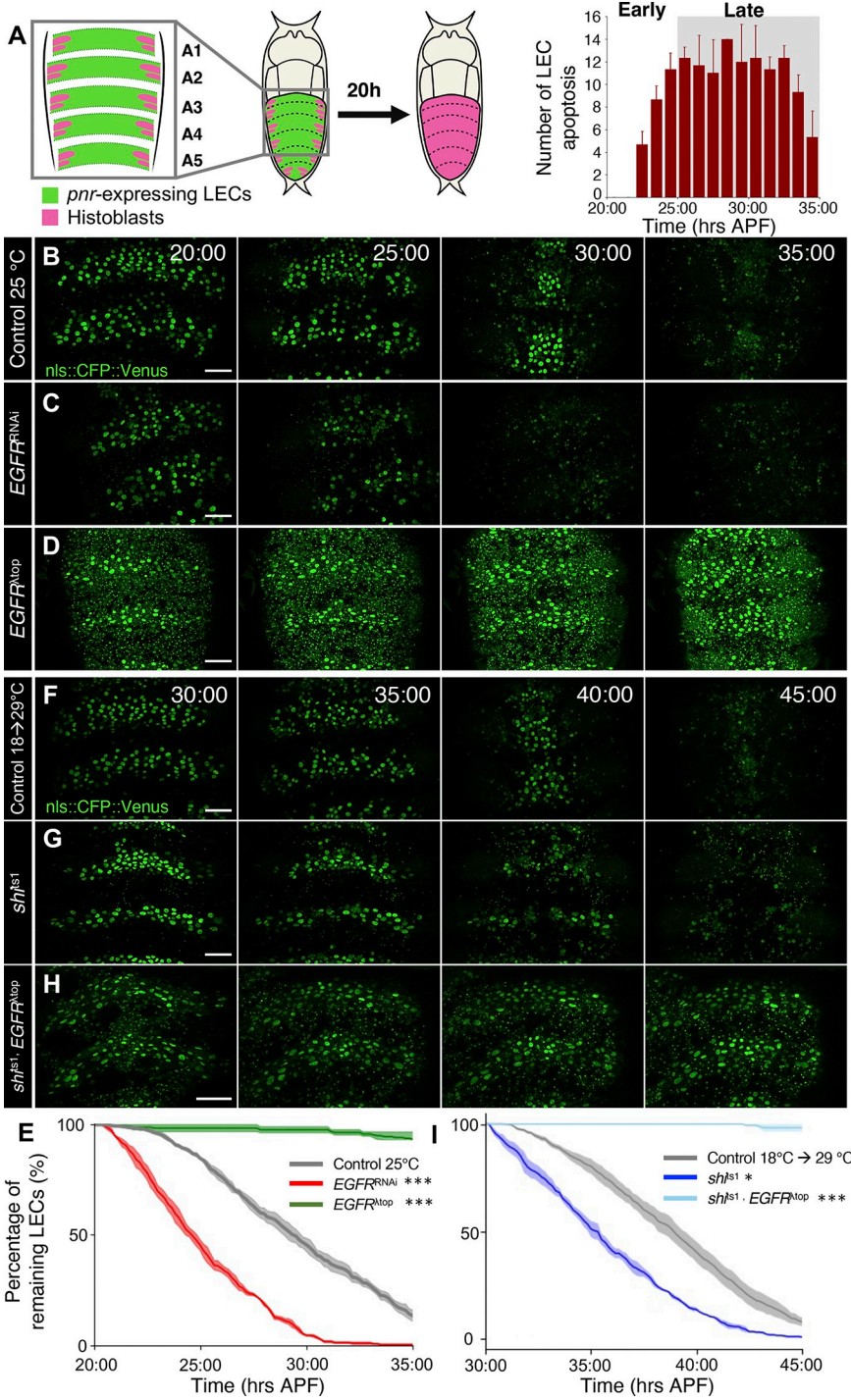

**Fig 1. EGFR signaling regulates LEC elimination rate and acts downstream of endocytic activity.** (A) Diagrams of: Left, *pnr* expression pattern (green) and diagram of LEC (green) elimination and replacement by histoblasts (magenta) within 20 h. Right, LEC elimination numbers at the specified time points (*N* = 3 pupae). (B–D) LEC elimination process in control (B), *EGFR*^RNAi (C), and *EGFR*^λtop (D) expressing pupae. Time stamp indicates hAPF (hrs:min). Scale bars 50 μm. (E) Percentage of remaining LECs (number of cells at t hAPF / number of cells at 20 hAPF). (F–H) LEC elimination process in control at 18˚C (F), *shi*^TS (G), and *shi*^TS plus *EGFR*^λtop (H) expressing pupae. Scale bars 50 μm. (I) Percentage of remaining LECs (number of cells at t hAPF / number of cells at 20 hAPF). *n* = 2 segments of 3 pupae. Error bars are SEM. Kolmogorov–Smirnov test, *$P < 0.05$, **$P < 0.01$, ***$P < 0.001$. Genotypes: (B, F) *ywhsFlp*/+; UAS-nls::CFP::Venus/+; *pnr*-GAL4/+. (C) *ywhsFlp*/+; UAS-nls::CFP::Venus/+; *pnr*-GAL4/UAS-*EGFR*^RNAi. (D) *ywhsFlp*/+; UAS- nls::CFP::Venus /+; *pnr*-GAL4/ UAS-EGFR ^λtop. (G) *ywhsFlp*/+; nls::CFP::Venus /+; UAS-

*shi*$^{TS}$, *pnr*-GAL4/+. (H) *ywhsFlp*/+; nls::CFP::Venus, /+; UAS-*shi*$^{TS}$, *pnr*-GAL4/UAS-EGFR$^{\lambda top}$. The data underlying the graphs shown in the figure can be found in https://zenodo.org/records/13290047. EGFR, epidermal growth factor receptor; hAPF, hours after puparium formation; LEC, larval epidermal cell.

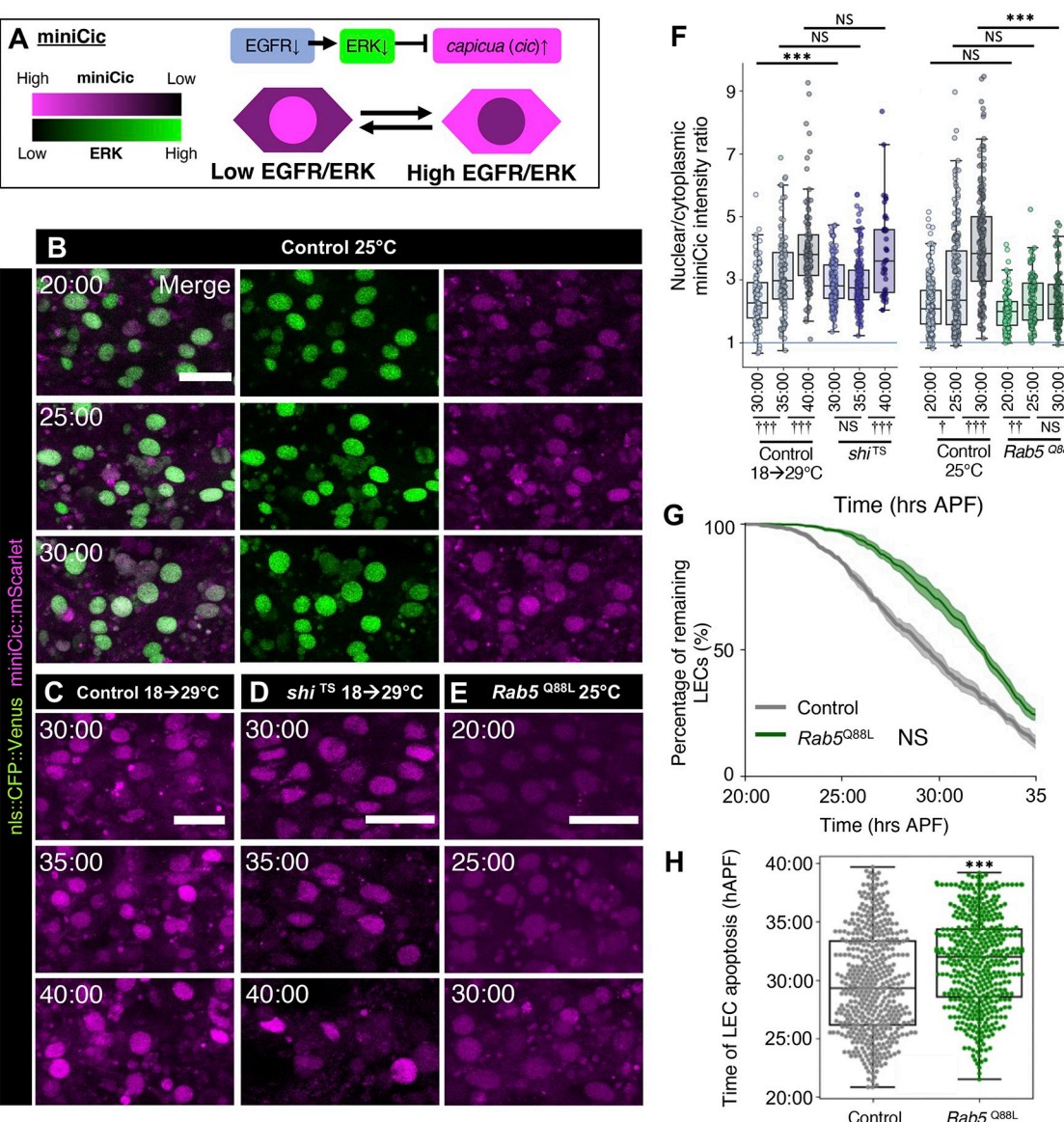

**Fig 2. EGFR/ERK signaling activity decreases throughout LEC elimination with endocytic activity dependency.** (A) Schematic of miniCic construct function. (B–E) miniCic::mScarlet and nls::CFP::Venus of LECs at the specified time points for 25˚C control (B), 18→29˚C control (C), *shi*$^{TS}$ (D), *Rab5*$^{Q88L}$ (E) pupae. Time stamps indicate hAPF (hrs:min). Scale bars 50 μm. (F) Nuclear / cytoplasmic ratio of miniCic in LECs from (B–E). *n* = 50 cells each from 3 pupae. Mann–Whitney test, NS: not significant, vs. controls ***$P < 0.001$, vs. previous time point †$P < 0.05$, ††$P < 0.01$, †††$P < 0.001$. (G) Percentage of remaining LECs for 25˚C control vs. *Rab5*$^{Q88L}$. *n* = 3 pupae for control and 5 pupae for *Rab5*$^{Q88L}$. Error bars are SEM. Kolmogorov–Smirnov test. NS = not significant, $P = 0.52$. (H) Swarmplot of LEC apoptosis counts at each time point for 25˚C control vs. *Rab5*$^{Q88L}$. Each data point indicates 1 cell. *N* = 3–5 pupae. Mann–Whitney test, ***$P < 0.001$. Genotypes: (B, C) *ywhsFlp*/+; *tubP*-miniCic::mScarlet /+; UAS-nls::CFP::Venus, *pnr*-GAL4/+. (D) *ywhsFlp*/+; *tubP*-miniCic::mScarlet /+; UAS-nls::CFP::Venus, *pnr*-GAL4/ UAS-*shi*$^{TS}$. (E) *ywhsFlp*/+; *tubP*-miniCic::mScarlet /+; UAS-nls::CFP::Venus, *pnr*-GAL4/ UAS-*Rab5*$^{Q88L}$. The data underlying the graphs shown in the figure can be found in https://zenodo.org/records/13290047. EGFR, epidermal growth factor receptor; ERK, extracellular-signal regulated kinase; hAPF, hours after puparium formation; LEC, larval epidermal cell.

decrease in endocytic activity in LECs, which was reported in our previous work [14]. While the reduction of endocytic activity ceases around 26 hAPF during the transition into the late phase [14], we observed that ERK activity continuously decreases even during the late phase. To show that the decrease of average ERK activity is caused by the reduction in endocytic activity, we analyzed ERK activity in the background of *shi*^TS. Abolishing endocytic activity by *shi*^TS decreased average ERK activity, which is limited in the early phase (Fig 2D and 2F). Consistently, we also found that over-activation of endocytic activity by expressing a constitutively active form of the endosome internalization regulator Rab5 GTPase (Rab5 ^Q88L) resulted in the maintenance of high average ERK activity compared to control in the late phase [36,37] (Fig 2E and 2F). Consequently, Rab5 ^Q88L significantly delayed LEC elimination compared to control (Fig 2G and 2H). These results indicate that ERK activity globally decreases over time and that the decrease of ERK is regulated by the reduction of endocytic activity during the LEC elimination period.

## The EGFR/ERK signaling fluctuates during LEC elimination depending on the initiation of apoptosis

Observing the temporal evolution of ERK activity in LECs during the elimination period, we noticed that accompanying the global down-regulation, ERK activity also fluctuates differently for each cell (Fig 3A, left). In order to obtain temporal profiles of the miniCic intensity for individual LECs, we automated the signal intensity measurements for the miniCic signal by combining with the tracking of nuclei labeled with a nuclear localized CFP::Venus protein. We found that individual LECs undergo multiple EGFR/ERK up-regulation and down-regulation events in pulses throughout the LEC elimination period, to a lesser extent in the late phase (Fig 3A, left). This observation reminded us of the ERK pulses in other models such as the *Drosophila* pupal notum and cell culture [30,38]. We therefore considered that there may be additional dynamics of ERK signaling during LEC elimination. To investigate whether these additional ERK dynamics are also regulated by endocytic activity, we analyzed the temporal evolution of miniCic nuclear intensity in the background of *shi*^TS and found that ERK activity fluctuations were diminished in *shi*^TS-expressing LECs as well as *EGFR* ^RNAi (Fig 3A, right). We further analyzed the intensity of these transient ERK up-regulations by measuring the average decrease of miniCic signal intensities in each ERK pulse (Fig 3B, top, see Materials and methods for details). As expected, the degrees of ERK up-regulation events were significantly diminished in *EGFR* ^RNAi and *shi* ^TS-expressing LECs (Fig 3B, bottom). These results suggest that endocytic activity in LECs regulates not only global ERK activity but also an additional mechanism that induces the fluctuation of ERK activity.

We next examined if the fluctuation of ERK is caused by caspase activation. We sought to block LEC elimination by expressing a dsRNA of the trio of *Drosophila* pro-apoptotic genes *reaper*, *hid*, and *grim*, collectively called RHG (Fig 3C–3E). Indeed, this knock-down of RHG completely inhibited LEC elimination but not the midline migration of LECs during the late phase, similar to our description of LEC dynamics upon overexpression of the apoptosis inhibitor *Diap1* [14]. We observed that miniCic signals were more homogenous in all LECs, with minimal fluctuations, suggesting that the fluctuation of ERK is caused by pro-apoptotic genes (Fig 3F and 3G). Interestingly, the average miniCic intensity started to increase at around 30 h APF when LECs further migrate towards the midline (Fig 3H), suggesting that the global ERK activity decrease is independent of pro-apoptotic gene activity. While these data suggest that the global basal ERK activity is regulated by a yet-unknown mechanism, we cannot still rule out a possibility that a low-level caspase activity down-regulates the global ERK activity.

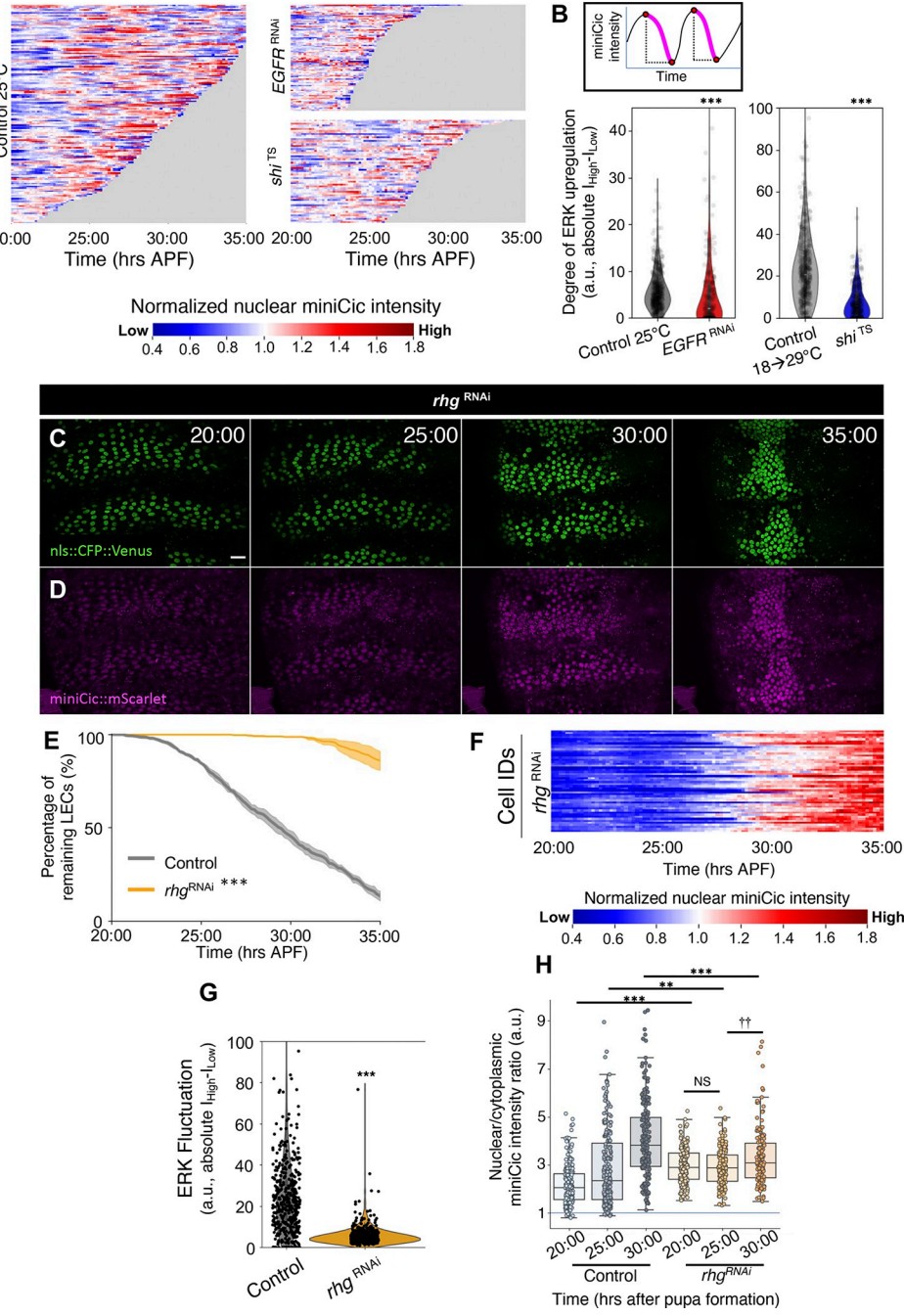

**Fig 3. EGFR activity in individual LECs fluctuates, which requires initiation of apoptosis.** (A) Heatmaps of nuclear miniCic::mScarlet intensity over time for a subset of LECs in all control (left), *EGFR*^RNAi (right top), and *shi*^TS (right bottom) pupae. Each line represents 1 cell. (B) Schematic of the time periods labeled in magenta, measured for the negative change in miniCic intensity (ERK up-regulation) (top). Violin plots of the degree of ERK up-regulation in change of absolute intensity for controls, *EGFR*^RNAi, and *shi*^TS LECs (bottom). Each dot represents the average up-regulation for each cell. $N$ = 3–4 pupae for each genotype. Error bars are SEM. Mann–Whitney test, ***$P$ < 0.001. (C, D) nls::CFP::Venus signals (C) and miniCic (D) in *rhg*^RNAi expressing LECs the specified time points APF (hrs:min). Scale bar 50 µm. (E) Percentage of remaining LECs over time. $n$ = 3 pupae. (F) Heatmap of the miniCic intensity of a representative *rhg*^RNAi pupa normalized to individual cell average. (G) Violin plots of the degree of ERK up-regulation in control and *rhg*^RNAi LECs in change of absolute intensity. $N$ = 3–4 pupae for each genotype. Error bars are SEM. Mann–Whitney test, ***$P$ < 0.001. (H) Nuclear / cytoplasmic ratio of miniCic in LECs. $n$ = 50 cells / pupa, 3 pupae. Error bars are SEM. Mann–Whitney test, **$P$ < 0.01 ***$P$ < 0.001. Genotypes: (A, B) *ywhsFlp*/+; *tubP*-miniCic:: mScarlet /+; UAS-nls::CFP::Venus, *pnr*-GAL4/+. *ywhsFlp*/+; *tubP*-miniCic::mScarlet /+; UAS-nls::CFP::Venus, *pnr*-

GAL4/ UAS-*EGFR*[RNAi]. *ywhsFlp*/+; *tubP*-miniCic::mScarlet/+; UAS-nls::CFP::Venus, *pnr*-GAL4/ UAS-*shi*[TS]. (C–H)
*ywhsFlp*/+; *tubP*-miniCic::mScarlet/+; UAS-nls::CFP::Venus, *pnr*-GAL4/+. *ywhsFlp*/+; *tubP*-miniCic::mScarlet /+;
UAS-nls::CFP::Venus, *pnr*-GAL4/UAS-*rhg*[RNAi]. The data underlying the graphs shown in the figure can be found in
https://zenodo.org/records/13290047. EGFR, epidermal growth factor receptor; ERK, extracellular-signal regulated
kinase; LEC, larval epidermal cell.

## LECs neighboring a primary apoptotic LEC exhibit a transient increase in ERK signaling activity

Previous reports observed the up-regulation of ERK in clusters of cells in response to apoptotic events [30,38–41]. In support of this notion, our data suggest that the fluctuations of ERK activity are suppressed by the inhibition of LEC apoptosis (Fig 3C–3H). In addition, we confirmed that the down-regulation of ERK precedes caspase activation and nuclear breakdown in individual LECs as previously reported, suggesting that the observed ERK activity fluctuation caused by pro-apoptotic genes is unlikely to be a cell autonomous effect (Figs 3C–3H and S2) [29,30,38]. To test whether the ERK activity in cell clusters respond to apoptotic events in a non-cell autonomous manner, we examined the dynamics of ERK activity in LECs neighboring an apoptotic LEC. We identified apoptotic LECs using nuclear breakdown as a signature for the end of apoptosis and measured the EGFR activity of LECs in the local vicinity of the apoptotic LECs (Figs 4E, S2E, and S2F). During the early phase, concurrent with LEC apoptosis, we observed a transient increase of ERK, represented by the transient decrease of miniCic, in several LECs neighboring the primary apoptotic LEC preceding apoptosis by about 60 min (Fig 4A, 4F, 4J and S1 Movie). This ERK pulse initiates during the gradual decrease of ERK and around the start of the apoptotic caspase activation (S2D Fig). In contrast, during the late phase, this transient ERK increase was only scarcely observed (Fig 4B, 4G, 4J and S1 Movie), which is in support of the less fluctuations seen at the late phase from the heatmaps (Fig 3A, left). Careful observations revealed that in this low EGFR activity, small clusters of 3 or more cells undergo concurrent apoptotic events, which we determined by the near-simultaneous nucleus breakdown of neighboring cells. There are a few exceptional cases where multiple instances of apoptosis occur in a cluster within a little prolonged period (around 30 min) of the primary apoptosis (Fig 4B' and 4G, yellow line). From these observations, we reasoned that the dynamically fluctuating ERK activities, which are represented in the heatmaps (Fig 3A), mainly originate from the transient increase of ERK to induce the prolonged survival of LECs neighboring primary apoptotic LECs. Furthermore, endocytic activity may be also involved in this process, especially during the early phase but not the late phase (Fig 4E). To test this idea, we genetically inhibited EGFR or endocytic activity and found that the absence of EGFR or endocytic activity abolished the transient increase of ERK activity (Fig 4C, 4D, 4H–4J and S2 Movie). Similar to the late phase in control pupae, we observed that even during the early phase, several cells in LECs exhibit concurrent decrease in ERK activity before their clustered apoptosis. Together, these results suggest that the dynamic fluctuations in ERK activity in LECs are the results of transient increase in cell clusters. The transient increase in cell clusters may potentially prevent excessive apoptosis among adjacent cells in a short period.

## Induction of apoptosis is sufficient to trigger the transient ERK activation in neighbor LECs

To examine whether LEC apoptosis is sufficient to trigger transient ERK activity in neighboring cells, we optogenetically induced apoptosis in some LECs while observing the ERK activity in neighboring cells. Here, we utilized OptoDRONC, an optogenetic tool that triggers caspase activation (Figs 5C, S3A, and S3B). Due to technical limitations, we selected settings that

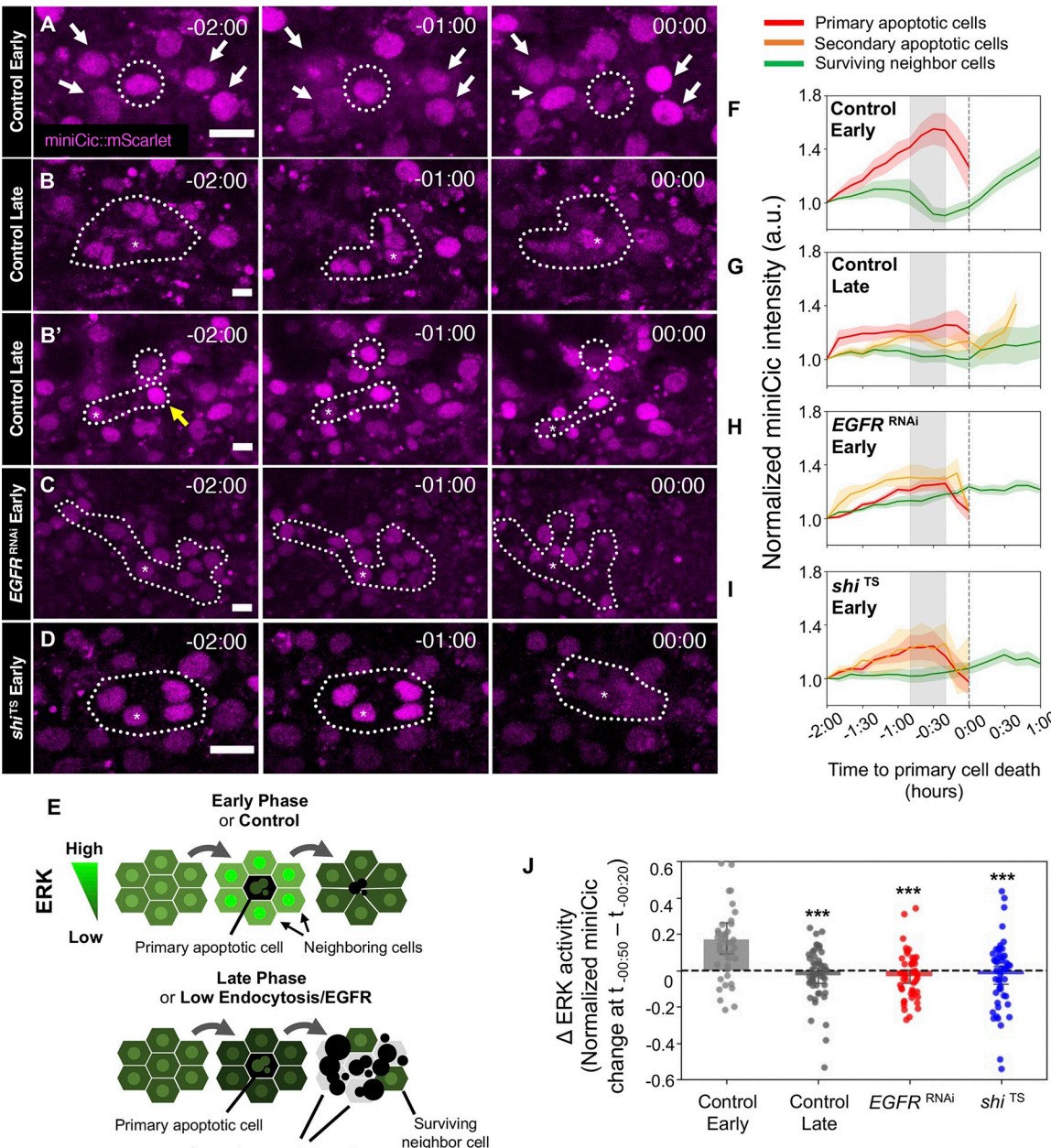

**Fig 4. ERK is activated in LECs neighboring a primary apoptotic LEC.** (A–D) miniCic::mScarlet in (A) early control, (B, B') late control, (C) *EGFR*^RNAi, and (D) *shi*^TS at the specified times (hrs:mins) before apoptosis. (B') A cluster in late control with non-concurrent, but shortly delayed apoptosis (30 min) marked with yellow arrows. Dotted circles indicate apoptotic cells. White arrows indicate neighbor cells exhibiting transient ERK up-regulation. Asterisks mark the primary apoptotic cell used in analyses. Scale bars: 20 μm. (E) Summary schematics for transient ERK activity in LECs in the indicated conditions (phase, EGFR, or endocytic levels) observed in (A–D), with cell role designations used in the study. (F–I) Pooled graph of miniCic intensity of LECs for each corresponding genotype. Red: primary apoptotic LECs, Orange: secondary/concurrent apoptotic LECs, Green: surviving neighboring LECs. Gray area indicates time period of ERK up-regulation events (50–20 min before apoptosis / $t_{-50}$ –$t_{-20}$). (J) Change of miniCic signal intensity during the time period of ERK up-regulation ($t_{-50}$ –$t_{-20}$). $n$ = 3–4 LEC clusters (each cluster: 1 primary, 4–7 neighbors) / pupa, 3 pupae. Error bars are SEM. Mann–Whitney test vs. Control Early, ***$P < 0.001$. Genotypes: (A–D, F–J) *ywhsFlp*/+; *tubP*-miniCic::mScarlet /+; UAS-nls::CFP::Venus, *pnr*-GAL4/+. *ywhsFlp*/+; *tubP*-miniCic::mScarlet /+; UAS-nls::CFP::Venus, *pnr*-GAL4/ UAS-*EGFR*^RNAi. *ywhsFlp*/+; *tubP*-miniCic::mScarlet /+; UAS-nls::CFP::Venus, *pnr*-GAL4/ UAS-*shi*^TS. The data underlying the graphs shown in the figure can be found in https://zenodo.org/records/13290047. EGFR, epidermal growth factor receptor; ERK, extracellular-signal regulated kinase; LEC, larval epidermal cell.

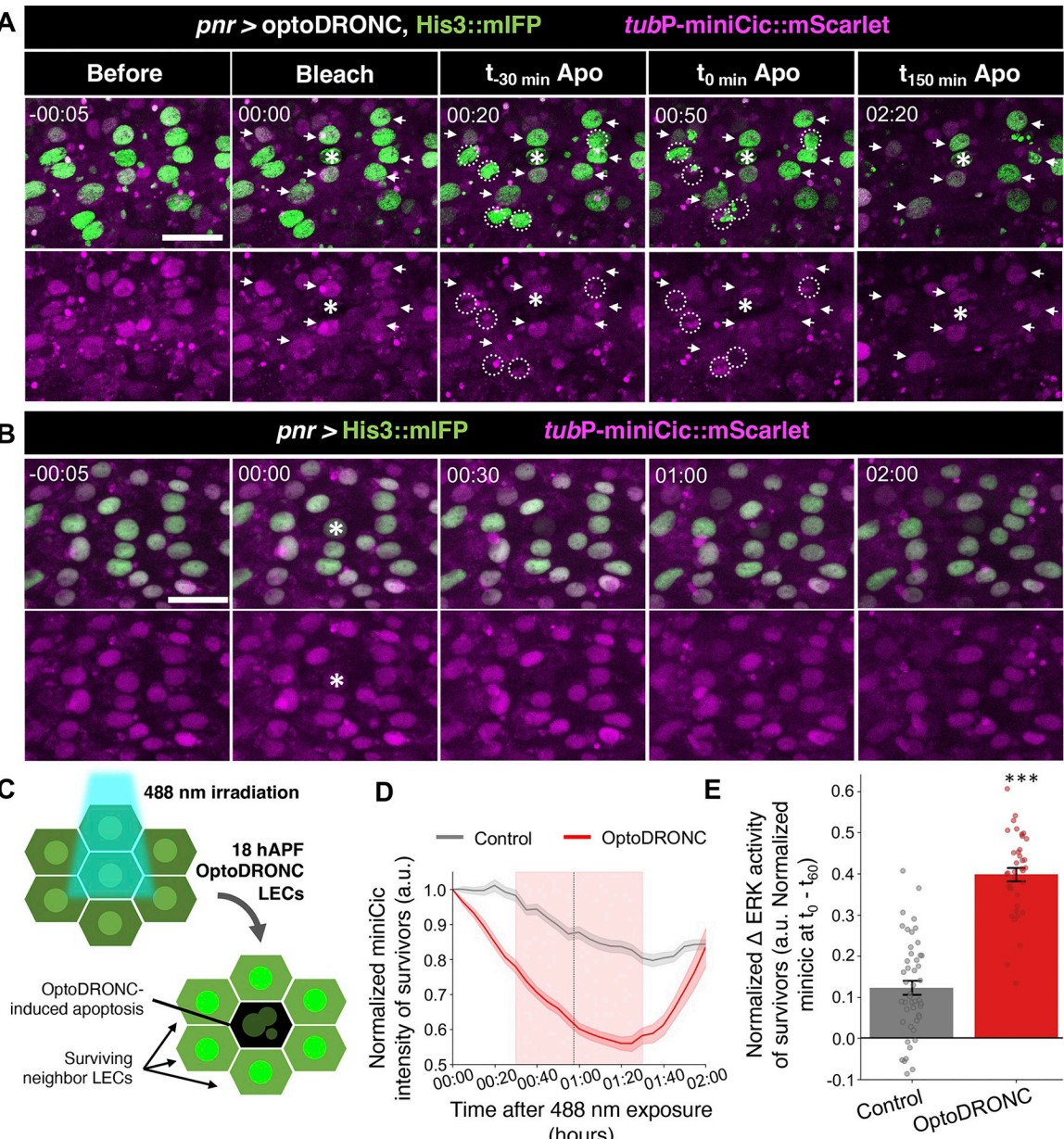

**Fig 5. Artificially inducing LEC apoptosis promotes ERK up-regulation of neighbor cells.** (A, B) Confocal images of *tubP*-miniCic::mScarlet and His3::miRFP for OptoDRONC-expressing (A) or control (B) LECs upon mild 488 nm laser exposure. Time stamp indicates time to exposure (hrs:min). Asterisk (*) marks the exposure point, dotted ellipses indicate OptoDRONC induced cell deaths. White arrows point the surviving neighbor cells with transient ERK activation. Scale bars: 50 μm. (C) Schematic of the analysis workflow for the experiments represented in A and B. Only surviving LECs were analyzed for EGFR activity in 488 nm exposed OptoDronc LECs, and all LECs for control pupae, as there are no apoptosis. (D) Normalized miniCic intensity of miniCic in surviving LECs from surrounding OptoDRONC-induced apoptotic cells. Pink shaded area indicates the time range of OptoDRONC-induced apoptosis, and the dotted vertical line is the mean time of apoptosis. Control $n = 3$ LEC clusters, each cluster contains 4–7 surviving cells. Control $n = 4$ LEC clusters, each cluster contains 4–7 surviving cells. (E) Increase of EGFR activity (intensity at the start of exposure ($t0$)–intensity at the mean time of OptoDRONC-induced apoptosis ($t60$)). Time stamps in A and B indicate the time from 488 nm exposure (hrs:min). Error bars are SEM. Mann–Whitney test, ***$P < 0.001$. Genotypes: (A, B, D, E) *ywhsFlp*/+; *tubP*-miniCic::mScarlet, UAS-OptoDRONC/+; UAS-His3::mIFP, *pnr*-GAL4/+. *ywhsFlp*/+; *tubP*-miniCic::mScarlet/+; UAS-His3::mIFP, *pnr*-GAL4/+. The data underlying the graphs shown in the figure can be found in https://zenodo.org/records/13290047. EGFR, epidermal growth factor receptor; ERK, extracellular-signal regulated kinase; LEC, larval epidermal cell.

allowed us to consistently promote OptoDRONC-induced apoptosis of several cells (approximately 100 μm away at maximum) but not of single cells, where the exact timing of apoptosis varies between 30 and 90 min after exposure between samples (Fig 5A, 5D, and 5E and S3 Movie, also see Materials and methods). The activation of OptoDRONC-induced transient ERK activation that returned to lower activity shortly afterward (20 to 30 min after nuclear breakdown of primary apoptotic LECs) in the surviving surrounding cells. Interestingly, we observed that ERK up-regulation in the LECs that underwent the induced apoptosis did not arrest the apoptosis process, which confirms (i) the downstream role of caspases to ERK; and (ii) that these apoptotic LECs were induced by OptoDRONC activation. In contrast, ERK activation was not induced by 488 nm laser in the LECs of early phase pupae without Opto-DRONC (Fig 5B, 5D, and 5E and S4 Movie). Our results from the optogenetic experiments showed that LEC apoptosis is sufficient to induce transient ERK up-regulation events in neighboring LECs.

## Down-regulation of endocytic and ERK activity increase the frequency of clustered LEC elimination

Apoptosis triggers the survival of neighboring cells by transiently up-regulating EGFR activity in neighboring cells in other systems [30,38]. We therefore examined if the loss of endocytosis/EGFR signaling spatiotemporally changes LEC elimination patterns. Analysis of the timing and location of LEC elimination over time in control pupae revealed that initially, LECs undergo apoptosis in an isolated, single-cell manner (Fig 6A and 6E). However, approaching and during the late phase, LECs tended to die concurrently as clusters of 3 or more cells (Fig 6B and 6E), suggesting a switching of the mode from isolated single cell to clustered cell elimination over the transition from early to late phase. In contrast, inhibiting ERK signaling or endocytic activity increased the frequency of clustered LEC elimination even when the early phase initiated, skipping the phase with single-cell eliminations (Figs 6C–6E, 6G, and S4B). We observed that the clusters form in lines, clusters, or a combination of lines and clusters containing varying numbers of cells (S5A Fig). The inhibition of ERK signaling or endocytic activity promoted the occurrences of larger clusters—some reaching more than 10 cells in single clusters (S5B Fig). These observations led us to hypothesize that clustered apoptosis is a unique mode of rapid cell removal, spatiotemporally regulated by the modulation of endocytic activity and ERK signaling. However, it is still possible that clustered elimination is a mere consequence of high elimination rate and increased fraction of dying cells in the late phase. Therefore, we compared the frequency of clustered cell elimination in tissues and that of random processes by implementing in silico simulations. The simulations were performed using the initial cell positions and elimination rate from our experimental control, EGFR [RNAi], and shi [TS] pupae, in fully randomized spatial patterns of cell elimination (Figs 6F and S4A and S5 Movie, see Materials and methods). These simulations revealed that fully stochastic order of apoptosis, despite having the elimination rates mimicking the in vivo experiments, hardly yields clustered elimination (Fig 6F and S5 Movie). These results support our hypothesis that clustered apoptosis events are unlikely to be caused by random chances. Combining our results, the absence of apoptosis-induced ERK up-regulation and the frequent cluster elimination in the late phase suggest that the loss of transient ERK activity promotes clustered cell death. Consistent with this idea, ERK activity is highly correlated with cluster apoptosis frequency ($r = 0.937$, $P = 0.00577$, Fig 6H). The lower the average ERK activity was, the more clustered eliminations were observed throughout all experiments irrespective of developmental time, supporting the idea that ERK activity protects cells from mass apoptosis in clusters. We note that there is a minor increase of clustered elimination in the 18 to 29°C shifted

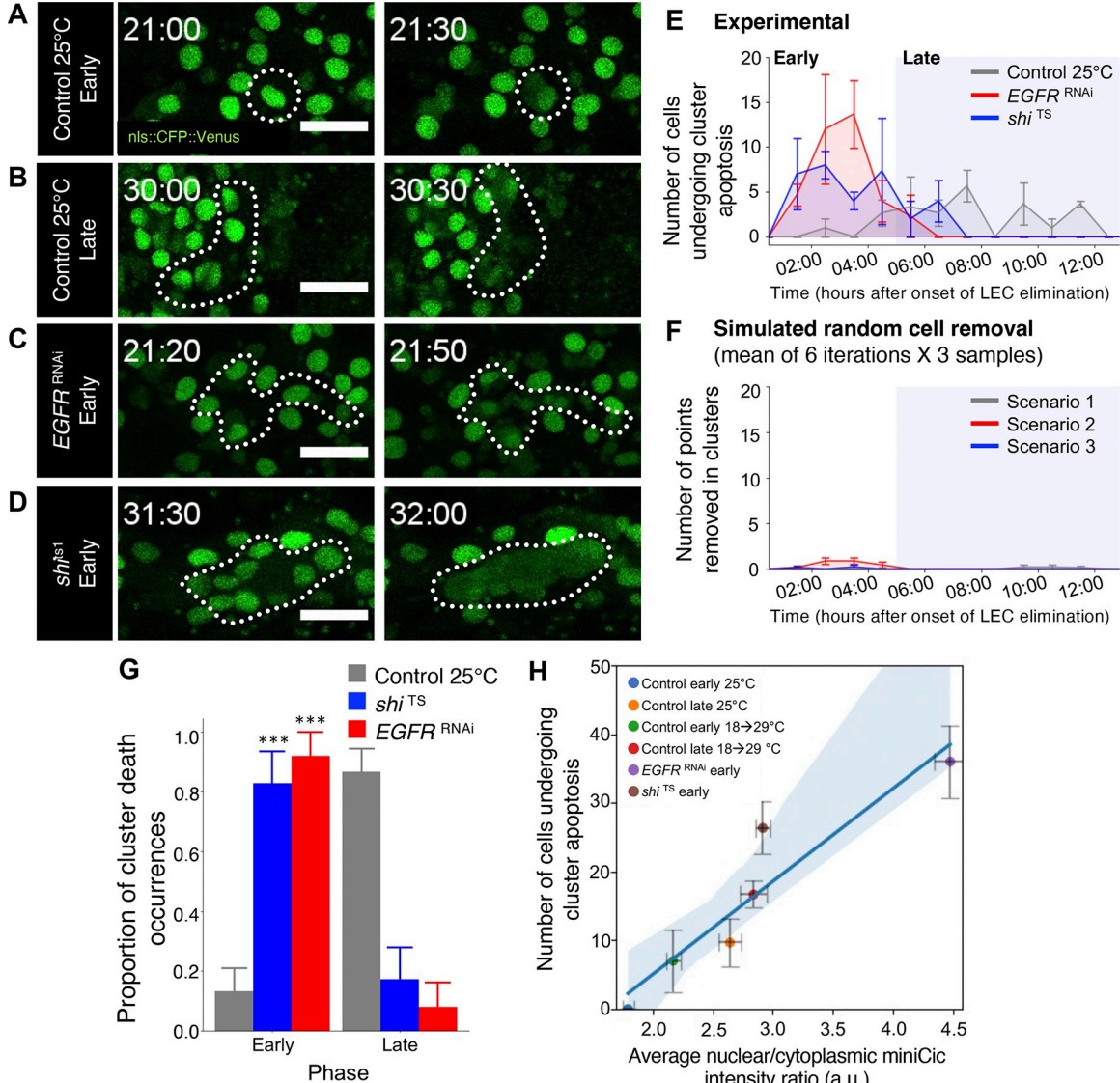

**Fig 6. Low ERK signaling promotes clustered LEC apoptosis.** (A–D) nls::CFP::Venus in early control (A), late control (B), *EGFR*^RNAi (C), and *shi*^TS (D) before and during cell death events. Time stamp indicates hAPF (hrs:min). Dotted circle indicates apoptotic cells. Scale bar: 50 μm. (E) Average experimental number of LECs dying in clusters for every hour in control, *EGFR*^RNAi and *shi*^TS pupae. (F) Average number of LECs dying in clusters for every hour in random cell removal simulations (6 iterations per sample) for the same initial cell positions and the same cell elimination rates as control (Scenario 1), *EGFR*^RNAi (Scenario 2), and *shi*^ts1 (Scenario 3) pupae. Three samples for each genotype for a total of 18 iterations for each scenario. (G) Average proportion of clustered cell deaths in the early and late phases for the indicated pupae. *n* = 3 pupae each. Error bars are SEM. Mann–Whitney test, ***$P < 0.001$. (H) Correlation between global average nuclear/cytoplasmic miniCic intensity in LECs and the number of cluster cell deaths in the early or late phases of the indicated pupae. Each data point represents the average values of the pupae in the indicated conditions. *N* = 3 pupae each. Error bars are SEM. Pearson's correlation coefficient (r): 0.937, *P* = 0.006. Genotypes: (A–D, F–H) *ywhsFlp*/+; *tubP*-miniCic::mScarlet /+; UAS-nls::CFP::Venus, *pnr*-GAL4/+. *ywhsFlp*/+; *tubP*-miniCic::mScarlet /+; UAS-nls::CFP::Venus, *pnr*-GAL4/ UAS-*EGFR*^RNAi. *ywhsFlp*/+; *tubP*-miniCic::mScarlet /+; UAS-nls::CFP::Venus, *pnr*-GAL4/ UAS-*shi*^TS. The data underlying the graphs shown in the figure can be found in https://zenodo.org/records/13290047. ERK, extracellular-signal regulated kinase; hAPF, hours after puparium formation; LEC, larval epidermal cell.

control LECs compared to the 25°C LECs, which we consider to be caused by the minor increase on the cell elimination rate at higher incubation temperatures (Fig 1E and 1H). Overall, our results suggest that transient ERK up-regulation in cells surrounding apoptotic cells

prevents clustered LEC elimination in the early phase and the loss of transient ERK up-regulation induces clustered LEC apoptosis, which is associated with the acceleration of cell elimination rate in the late phase.

## Knock-down of the Drosophila EGFR ligand vein diminishes ERK dynamics and increases clustered apoptosis of LECs

Finally, we turned our attention to identifying the mechanism that triggers transient ERK activation and consequently prevents cluster cell elimination. Recent work in mammalian cell culture systems showed that ERK waves are dependent on an EGFR ligand, amphiregulin, that is released from apoptotic cells [42]. We therefore examined whether EGFR ligands trigger transient ERK activation in neighbor cells upon primary cell apoptosis. EGFR is activated when it binds with the ligands *spitz* (*spi*), *vein* (*vn*), *gurken* (*grk*), or *Keren* (*Krn*) in a context-dependent manner [43–48]. Knock-downs of these ligands revealed that *vn* RNAi accelerated the LEC elimination rate to a similar extent as *EGFR* RNAi (Figs 7A–7D, S6A, and S6B and S6 Movie). *spi* RNAi also promoted LEC elimination rate but to a markedly lesser extent (S6A–S6C Fig). Neither *grk* nor *Krn* affected the LEC elimination rates (S6A and S6B Fig). Time course analysis of ERK activity in single cells showed that the fluctuations that reflect transient up-regulation in clusters (Fig 3A and 3B) are diminished, as they are in the inhibition of EGFR, suggesting the loss of transient up-regulation of ERK activity (Fig 7E). It should also be noted that miniCic::mScarlet fluorescence in *vn* RNAi flies was weak overall for unknown reasons (Fig 7B). Finally, the knock-down of *vn* and the loss of ERK fluctuations was accompanied by a higher percentage of clustered LEC elimination in the early phase, although not to the degree of the EGFR knock-down (Fig 7F and 7G). In summary, our results showed that *vn* regulates the transient ERK activation in LECs and prolongs their survival, suggesting that the transient up-regulation of ERK in LECs, which prevents clustered cell deaths and prolongs their survival during the early phase, is partly a ligand-dependent process.

## Discussion

Our previous work has shown that the reduction of endocytic activity advances LEC elimination during abdominal epithelium tissue remodeling in *Drosophila*. This study provides the molecular and cellular mechanism that mediates the LEC elimination downstream to endocytic activity reduction by characterizing the role of the cell survival regulator ERK signaling. Similar to endocytic activity, ERK signaling activity in LECs is down-regulated gradually throughout the tissue remodeling process. Additionally, ERK activity increases locally and transiently in the cells adjacent to an apoptotic cell, with only few exceptions. The local ERK activation prevents those cells from undergoing cell elimination together with the primary apoptotic cells. In later stages or conditions where endocytic activity or EGFR is genetically inhibited, LECs die in clusters in a caspase-dependent manner, suggesting that the acceleration of cell elimination in the later stage is linked to the switch from single-cell to clustered-cell apoptosis. In other words, the key action is not an active acceleration of cell elimination in the late phase but rather the suppression of rapid cell elimination in the early phase regulated by endocytic activity in LECs. We further identified that the EGFR ligand *vein* (*vn*) is a key modulator of the transient ERK up-regulation that promotes the survival of cells, thereby showing that the transient ERK activations is at least partly a ligand-dependent process.

Our results revealed that endocytic activity in LECs directly mediates ERK signaling, which is known to antagonize caspase-dependent apoptosis in various models [26,32,33,49–53]. EGFR internalization through endocytosis has been shown to promote downstream MAPK signaling to regulate cell survival and proliferation [27,54,55]. Additionally, endocytic activity

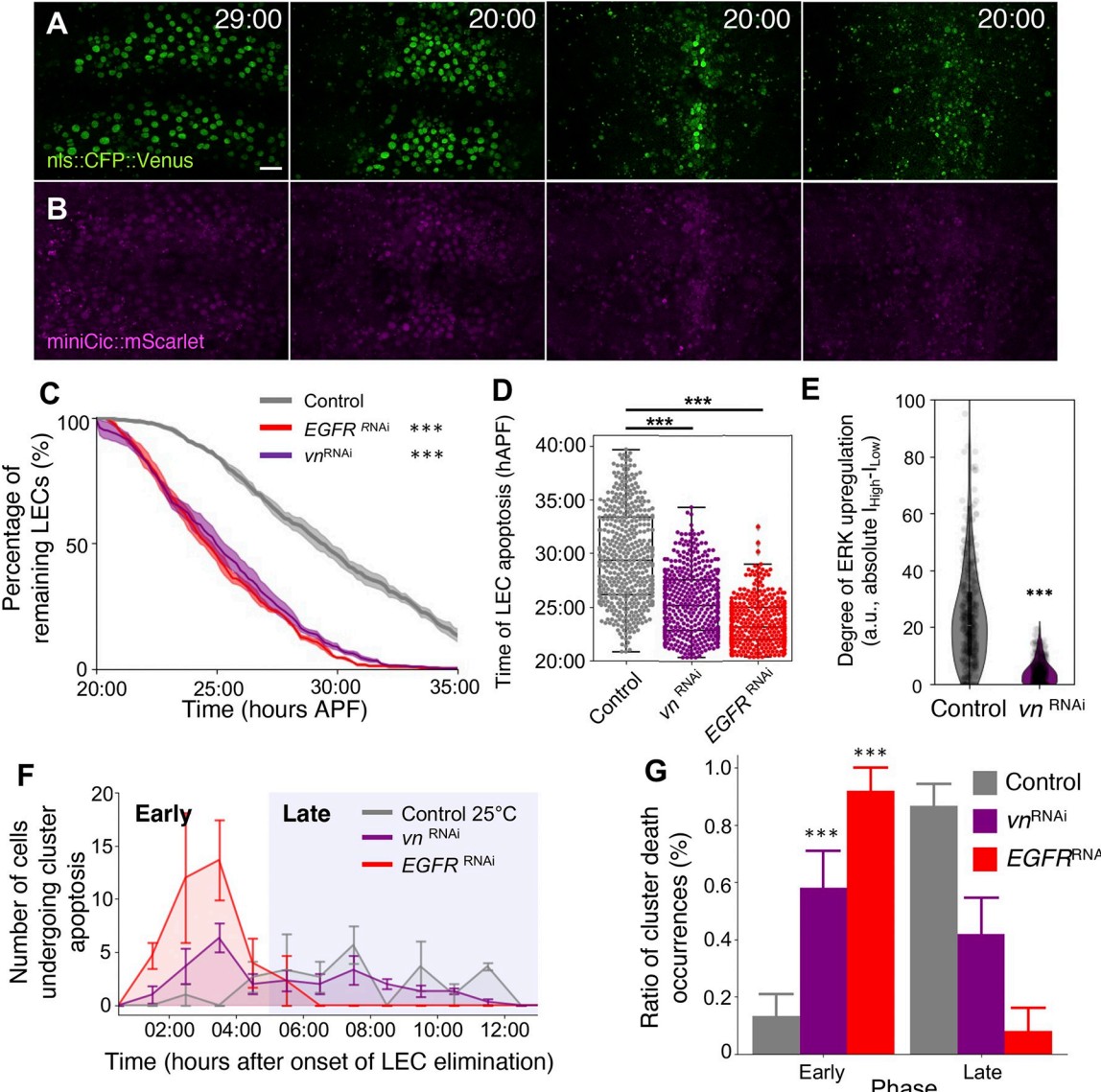

**Fig 7. Knockdown of the EGFR ligand *vein* diminishes ERK fluctuations.** (A, B) nls::CFP:Venus (A) and miniCic::mScarlet (B) in *vn*[RNAi] expressing flies at the specified times (hrs:mins) APF. Scale bar 50 μm. (C) Percentage of remaining LECs. *n* = LECs from 2 segments of 3 pupae each. Error bars are SEM. Kolmogorov–Smirnov test, ***$P < 0.001$. (D) Swarmplot of LEC apoptosis counts at each time point for 25˚C control vs. *EGFR*[RNAi] vs. *vn*[RNAi]. Each data point indicates 1 cell. *N* = 3–4 pupae. Mann–Whitney test, ***$P < 0.001$. (E) Violin plots of the degree of ERK up-regulation in control and *vn*[RNAi] LECs in change of absolute intensity. *N* = 3 pupae for each genotype. Error bars are SEM. Mann–Whitney test, ***$P < 0.001$. (F) Number of cells undergoing cluster cell eliminations in *vn*[RNAi] expressing flies at 1 h intervals. (G) Proportion of cluster cell deaths in early or late phases in control, *EGFR*[RNAi], and *vn*[RNAi] expression pupae. *n* = 3 pupae. Error bars are SEM. Mann–Whitney test, ***$P < 0.001$. Genotypes: *ywhsFlp*/+; *tubP*-miniCic::mScarlet /+; UAS-nls::CFP::Venus, *pnr*-GAL4/+. *ywhsFlp*/+; *tubP*-miniCic::mScarlet /+; UAS-nls::CFP::Venus, *pnr*-GAL4/ UAS-*EGFR*[RNAi]. *ywhsFlp*/+; *tubP*-miniCic::mScarlet /+; UAS-nls::CFP::Venus, *pnr*-GAL4/ UAS-*vn*[RNAi]. The data underlying the graphs shown in the figure can be found in https://zenodo.org/records/13290047. EGFR, epidermal growth factor receptor; ERK, extracellular-signal regulated kinase; LEC, larval epidermal cell.

has been reported to regulate EGFR's recycling capacity which ultimately maintains receptor availability [56,57]. With the decrease of endocytic activity, both EGFR activation and availability gradually lower over time, increasing the probability that cells will undergo apoptosis. How does endocytic activity decrease over time during the LEC elimination period? It is likely that the process is regulated by ecdysone signaling, which is a master regulator of the

transcription network for metamorphosis. In many other tissues, 20-hydroxyecdysone broadly activates the downstream gene regulatory networks that trigger histolysis in many larval tissues [58–63]. It would therefore be interesting to test the involvement of ecdysone signaling in regulating the global endocytic activity. In addition to the systemic regulation, the decrease of global endocytic activity might be under additional layers of regulations. Here, we propose 2 possible mechanisms. The first one is the mechanical regulation. Reduction of endocytic activity was previously reported in amnioserosa cells during *Drosophila* embryo dorsal closure as the membrane tension increases and cell volume decreases [64]. Interestingly, it has been shown that mechanical tension on cell junctions in LECs increases by 5- to 10-fold at 27 hAPF compared to 20 hAPF [17]. Therefore, it is possible that the reduction of endocytic activity in LECs in the late stage is also caused by the increase in cell junction tension globally throughout the tissue. Another potential regulation that could facilitate the decrease of endocytic activity is a positive feedback loop between the endocytic activity and caspase activations in individual cells. It has been shown in an in vitro study that an initiator caspase, caspase-8, and an effector caspase, caspase-3, cleave the adaptor protein-2 (AP2) and the clathrin heavy chain (CHC), respectively [65]. Both AP2 and CHC regulate endocytosis through the formation of clathrin-coated pits at the plasma membrane. In a similar mechanism, the endocytic activity in LECs may be down-regulated by caspase activations. Combined with our results as well as our previous study [14], this interaction is considered to form a positive feedback loop, further decreasing endocytic activity. Consequently, this reduction would accelerate the transition of the cell state. The proposed 2 mechanisms—the mechanical model and the positive feedback—are not mutually exclusive, and both systems may also be regulated under the control of ecdysone signaling. Thus, it is likely that the combination of both molecular and mechanical systems promotes this decrease of endocytic activity in LECs.

Studies in developing tissues have focused primarily on isolated single-cell delamination, which occurs stochastically [66–68]. By contrast, our study focuses on a massive cell removal process that is programmatically regulated throughout development. We revealed that this tightly regulated acceleration of cell elimination rate is not a mere consequence of the increased frequency of isolated single-cell apoptosis, but rather results from a switch in the mode of cell elimination toward simultaneous apoptosis of cells in local clusters. The comparison between experimental data and simulations clearly showed that the clustered cell elimination is also not a mere consequence of the increased cell elimination rate. Further improvements in our simulations may provide even better comparisons since our simulations do not consider the midline migration and other passive movements of cells due to apoptotic events which may influence the distribution of apoptosis. Despite its limitations, our study constitutes the first example of implementing clustered cell elimination in the context of development and demonstrates that it plays an active role in accommodating the proliferating cells of neighboring tissues—in our model, the *Drosophila* adult histoblasts—which increase exponentially and thereby replace the shrinking tissue more rapidly as tissue remodeling progresses. The role of clustered elimination in this context is distinct from the conventional understanding that clustered apoptosis is disadvantageous by compromising tissue integrity [30,38]. In fact, fluorescent labeling of cell boundaries showed that there is no visible impairment in epithelial sealing despite the presence of clustered apoptosis (S5C Fig). This observation contrasts with the formation of transient gaps between cells when cell deaths were artificially induced in the *Drosophila* pupal thorax [30]. The constant movement of both LECs and histoblasts may prevent the formation of transient gaps in the abdominal epidermis. Our study on the molecular mechanisms regulating LEC clustered apoptosis showed that transient ERK up-regulation is dependent on endocytic activity, which gradually decreases over the course of the LEC elimination process. As endocytic activity decreases, the LECs are prone to

undergo cluster cell elimination. It is possible that the abovementioned positive feedback mechanism and/or change in the mechanical environment play a role in the transition of the cell elimination mode from isolated to clustered apoptosis. In addition to this switch to the loss of local and transient ERK activation in the late phase, we have also identified the presence of a gradual global down-regulation of ERK signaling in LECs over the entire course of LEC elimination. A limitation of this study is in fact that there are no conditions to independently modulate and uncouple the basal ERK activity and the ERK pulses. While the global ERK down-regulation seems to reflect global down-regulation of endocytic activity in LECs, it is still unclear if the global down-regulation also contributes to the switch in cell elimination mode. Although the relative contributions of the basal ERK activity and the ERK pulses to the regulation of cluster apoptosis are unclear and still need to be further investigated, one possibility of the role of global down-regulation of ERK basal activity is that it may desensitize LECs to the transient ERK pulses and ensure cluster cell elimination occurs during later stages. The ERK pulses were reported to depend on frequency and intensity thresholds to successfully induce apoptosis-induced survival [38]. In this way, despite the continuous presence of ERK pulses, the low basal ERK activity in late phase LECs cause the ERK pulses unable to reach the required survival thresholds. Further investigations regarding the contributions of the apoptosis-induced ERK pulses and the basal ERK activity in protecting the neighboring cells from apoptosis and ultimately regulating the elimination rate will be interesting.

Transient, pulsed waves of ERK/Akt signals have been shown to be either ligand-dependent [38] or mechanically induced [29,30]. In other models, induced mechanical stretching promotes the growth and proliferation of both smooth muscle and alveolar epithelial cells via transient EGFR activation [69,70]. As exemplified by these studies, the regulation of EGFR activation by mechanical stretching has been well studied. Ligand-dependency of this mechanically gate ERK waves has been previously reported in MDCK cell collective migration [71]. Ligand-dependent ERK activity waves have been reported in *Drosophila* tracheal development [72]. The progressive expansion of ERK activation in the tracheal placode is regulated by the outward spreading of EGFR ligand source mediated by the spreading of the expression domain of *rho*, the gene encoding an activator of the *spi* ligand. HeLa cells under apoptotic stress were also reported to induce the non-autonomous survival of neighboring cells by secreting FGF2 growth factor in an MEF-ERK–dependent mechanism [40]. Our data further provided support for the idea that the transient ERK activation during apoptosis-induced survival of LECs is at least partly a ligand-dependent process. Interestingly, the transient up-regulation of ERK activity in LECs is dependent on *vn*, suggesting a distinct mechanism from that in the tracheal placode, although we cannot not rule out the involvement of *spi*. *vn* is a *Drosophila* homologue of mammalian neuroregulins (NRG), found to be expressed in a highly specific manner in developing ectoderm-derived tissues such as epithelial tissues and the nervous system as a constitutively active ligand without a transmembrane domain [31,73–76]. We speculate that apoptotic LECs release the *vn* needed for neighbor cell survival. Intriguingly, a recent report has shown that the extracellular vesicles derived from an extruding LEC are engulfed by neighbor cells [77], raising a possibility that the ligand is transferred through the extracellular vesicle which can be then internalized by neighbor cells in an endocytosis-dependent manner. However, our current work has only shown the effects of global down-regulation of *vn* to reduce the ERK fluctuations in LECs and has yet to demonstrate the dynamics of the Vn protein within the local LEC clusters during the transient up-regulation of ERK in LECs. Another potential mechanism is that Vn and to a minor degree Spi have a global permissive function where the presence of the ligand is prerequisite for preventing clustered cell elimination in the early phase independent of the regulation of the transient ERK activation. In this case, either: (1) the reduced ligand availability; or (2) the loss of EGFR availability and ligand-dependent activation

due to the loss of endocytic activity can cause the global down-regulation of the ERK signal to the level that cannot be overcome anymore by pulsed up-regulation of the ERK for cell survival. Lastly, while previous studies show that the ERK waves propagate multiple layers of cells [38], the ERK pulses in LECs are confined to only the cells adjacent to the primary apoptotic cells. In addition, all adjacent neighbor LECs do not necessarily undergo the ERK pulses. This local and inhomogeneous response may reflect the difference in receiving ligands or mechanical stress depending on their locations and geometry. Interestingly, a recent report showed that mechanostransduction of YAP signaling only in the largest neighbor cells induce compensatory proliferation in cell culture [78], suggesting a similar mechanism may play a role during LEC elimination. Understanding the precise mechanism by which ligands regulate cluster LEC elimination awaits further analyses.

Overall, this study revealed the 2 default states of cells depending on the availability of the ERK-dependent survival mechanism, where high ERK favors the isolation of single cell apoptosis and low ERK allows clustered cell elimination. Our findings indicate that the apoptosis-induced survival system is present even in tissues fated to undergo complete elimination in a narrow time window. It is likely that during the early phase, prior to histoblast dorsal expansion, ERK up-regulation increases the robustness of neighbor cells and prevents cluster cell deaths to sparsely allocate apoptosis and avoid large numbers of cell extrusions as a means to ultimately maintain epithelial structure, as suggested in previous reports [30,38,79]. In the late phase, by contrast, the loss of endocytic activity-dependent ERK up-regulation switches LECs to clustered apoptosis. This is likely performed to accommodate the space required by the accelerating proliferation of histoblasts. The spatiotemporal control of this system switches cell death behavior to pro-clustered death in order to meet the strict timing of tissue remodeling in *Drosophila* metamorphosis.

## Materials and methods

### Fly stocks

The Gal4/UAS system was utilized for tissue-specific expression. The tubP-miniCic::mScarlet and UAS-OptoDRONC flies were gifts from R. Levayer. The fly stocks used in this study are listed below (Table 1).

**Table 1. List of fly lines used in this study.**

| Genotype | Origin |
|---|---|
| UAS-nls::SCAT3 (nls::CFP::Venus) | [80] |
| *pnr*-Gal4 | [81] |
| *tubP*-miniCic::mScarlet | [29] |
| UAS-*EGFR*$^{RNAi}$ | Bloomington Stock Center #36773 |
| UAS-*EGFR*$^{\lambda top}$ | Bloomington Stock Center #59843 |
| UAS-*shi*$^{TS}$ | [35] |
| UAS-GC3Ai, *pnr*-GAL4 | Bloomington Stock Center #84319 |
| UAS-*rhg*$^{miRNA}$ | [82] |
| UAS-*Rab5*$^{Q88L}$ | Bloomington Stock Center #43335 |
| UAS-nls::mCherry | Bloomington Stock Center #38424 |
| UAS-OptoDRONC | [30] |
| UAS-*vn*$^{RNAi}$ | Bloomington Stock Center #67844 |
| UAS-*spi*$^{RNAi}$ | Bloomington Stock Center #34645 |
| UAS-*grk*$^{RNAi}$ | Bloomington Stock Center #55926 |
| UAS-*Krn*$^{RNAi}$ | Bloomington Stock Center #67558 |

## Live imaging

Prepupae (0 hAPF) were collected and kept at 25˚C for 20 h unless otherwise noted. For temperature sensitive experiments, pupae were incubated at 18˚C for 24 h then 29˚C for 6 h. Imaging starts at 20 hAPF for pupae incubated at 25˚C. Pupae undergoing temperature-shift (18˚C to 29˚C) were imaged starting from 30 hAPF to accommodate the developmental delay induced by 18˚C incubation. Prior to imaging, the pupae were washed in distilled water to clean remaining debris, and then immobilized on a stack of double-sided tape for dissection. Two-thirds of the pupal shell above the dorsal abdomen was removed to expose the underlying epithelium with a pair of dissecting forceps, then the sample was mounted on a cover glass. For 20 h, live imaging of LEC elimination was performed on a Leica TCS SP8 upright confocal microscope (Leica, Germany). The objective lenses used were: 20× NA 0.75 or 63× NA 1.40 oil-immersion objectives (Leica, Germany).

## Optogenetics

Male *w*; UAS-OptoDRONC, *tub*P-miniCic::mScarlet/CyO flies were crossed with female *ywhsflp*/+;; UAS-nls::mIRFP, *pnr*-GAL4/TM6B flies. F1 prepupae (0 hAPF) were collected and kept at 25˚C for 16 h, then dissected and mounted in dark room to avoid any apoptosis induced by either natural LEC elimination or OptoDRONC activity. Live imaging was performed on a Zeiss LSM980 with an Airyscan 2 inverted confocal microscope (Zeiss, Germany). The objective lens used was a Plan-Apochromat 20× NA 0.8 M27 (Zeiss, Germany). Clusters of cells were randomly selected from the channel with nuclear localization signal, focusing on 1 nucleus at the center as the target for laser exposure. An elliptical regions of interest (ROI) with 25 μm and 5 μm diameters was created, and 488 nm laser exposure was conducted 3 times at 5-min intervals of 15 s each. Image stacks were taken 5 min before and every 5 min for at least 2 h during and after exposure.

## Image and data analysis

All image analysis was performed in ImageJ 1.53t (National Institutes of Health, USA, URL: http://imagej.nih.gov/ij Java 1.8.0_202 (64-bit)).

**Automatic cell tracking and nuclear intensity measurements.** LECs in abdominal segments 2 and 3 with nls::CFP::Venus nuclear signals under the control of *pnr*-GAL4 driver were detected and tracked in all time frames using the TrackMate plugin in ImageJ [83]. Data for cluster elimination and nuclear miniCic intensities were obtained from the "Analysis" output of the plugin.

**MiniCic nuclear/cytoplasmic ratio analysis.** Small ROI, roughly 20 μm in diameter (matching the nuclear size) from the nuclei and cytoplasm directly adjacent to each nucleus were taken manually to measure the fluorescence intensities. The ratio values presented in the figures were obtained by dividing the nuclear intensity by the cytoplasmic intensity of the same cells.

**Manual analysis for nuclear miniCic intensity in cell clusters.** Each LEC cluster was selected by choosing an apoptotic cell at random. Nuclear miniCic intensities were measured between −120 to 0 min (0 indicates the time of primary cell apoptosis, marked by nuclear breakdown and signal diffusion) for primary and secondary apoptotic cells and −120 to 60 min for the surviving neighboring cells in each cluster.

For the optogenetic analysis, we included all the surviving cells neighboring the OptoDRONC-induced apoptotic LECs for the miniCic analysis.

**Cluster cell death detection.** LEC elimination data from TrackMate were preprocessed in R 4.0.2 (The R Foundation for Statistical Computing). These data were then used for cell

detection with a Python 3.9 code for neighbor detection with Delaunay Triangulation, and false positives for cell connections between second and third segments were removed by applying a maximum distance limit for detection. The loss of cell pairs at the same time or +1 frame (10-min interval as a margin for nuclear breakdown) was regarded as simultaneous cell death. If one cell ID of a pair was found in another pair of cells, the pairs were considered to be an interconnected pair. Interconnected pairs were visually inspected in the confocal images before being determined to be clustered cell deaths. Data presented contains only the detection for clustered cell deaths.

## Simulation for fully randomized cell death

Python 3.9 was used to build a code to randomize cell death order based on these criteria: (1) The initial positions of the cells are the XY positions at the first frame; and (2) cell death rate follows the experimental rates. All samples of the associated genotypes from the experiments were used as inputs for the simulation, each iterated for random cell removals for 6 times. For example, $EGFR^{RNAi}$ samples No. 1, 2, and 3 were each iterated for 6 times for a total of 18 outputs for each genotype. Each iteration of randomized cell deaths was then processed for cluster apoptosis detection as explained above. While our simulation model did not incorporate the LEC migration, the triangulation finds the shortest connections between points, which may mitigate the spatial limitations. The code is available at the following link: https://github.com/k0yuswantohoku/LEC_removal_simulation.

## Statistics

All statistical analysis and plot generation was performed in either R 4.0.2 (The R Foundation for Statistical Computing) or with Python 3.9 (The Python Software Foundation).

## Supporting information

**S1 Fig. miniCic reports EGFR signaling in LECs.** (A–C) Top row, nuclear signals and bottom row, miniCic signals in LECs at 20 hAPF in (A) control, (B) $EGFR^{RNAi}$, and (C) $EGFR^{\lambda top}$ expressing pupae. Scale bars: 50 μm. (D) Nuclear / cytoplasmic ratio of miniCic in LECs at 20 hAPF. $n$ = 50 cells / pupa, 3 pupae. (E) Normalized nuclear miniCic intensity in control LECs at 20, 25, and 30 hAPF. Error bars are SEM Mann–Whitney test, ***$P < 0.001$. Genotypes: (A) *ywhsFlp*/+; *tubP*-miniCic::mScarlet/+; UAS-nls::CFP::Venus, *pnr*-GAL4/+. (B) *ywhsFlp*/+; *tubP*-miniCic::mScarlet/+; UAS-nls::CFP::Venus, *pnr*-GAL4/ UAS-*EGFR*$^{RNAi}$. (C) *ywhsFlp*/+; *tubP*-miniCic::mScarlet/+; UAS-nls::CFP::Venus, *pnr*-GAL4/ UAS-*EGFR*$^{\lambda top}$. The data underlying the graphs shown in the figure can be found in https://zenodo.org/records/13290047. (PDF)

**S2 Fig. Down-regulation of EGFR precedes caspase activation in apoptotic LECs.** (A) Schematic of GC3Ai reporter of apoptotic caspase activity. (B) miniCic signals and (C) caspase activity in LECs in control pupae. White arrows indicate apoptotic LECs. Time indicates hours APF. Scale bars: (B) 50 μm and (C) 20 μm. (D) Plot of the time scale of normalized miniCic and GC3Ai intensity in LECs, aligned to the time of cell death. $n$ = 10 cells / pupa, 3 pupae. (E) Confocal images of nls::mCherry and GC3Ai in an apoptotic LEC. Time indicates hours to apoptosis. White arrowhead indicates ongoing nuclear breakdown. (F) Plot of the time scale of normalized GC3Ai intensity and nuclear sizes in LECs, aligned to the time of cell death. $n$ = 2–3 cells / pupa, 3 pupae. Scale bar: 20 μm. Errors are SEM. Genotypes: (B) *ywhsFlp*/+; *tubP*-miniCic::mScarlet/+; UAS-nls::CFP::Venus, *pnr*-GAL4/+. (C) *ywhsFlp*/+; +/+; *pnr*-GAL4, UAS-GC3Ai/+. (E) *ywhsFlp*/+; UAS-nls::mCherry/+; *pnr*-GAL4, UAS-GC3Ai/+. (E, F)

*ywhsFlp*/+; UAS-GC3Ai/+; *pnr*-GAL4/UAS-nls::mCherry. The data underlying the graphs shown in the figure can be found in https://zenodo.org/records/13290047.
(PDF)

**S3 Fig. OptoDRONC activation induces LEC deaths.** (A) Schematic of OptoDRONC (top) and workflow for optogenetically inducing OptoDRONC (bottom). (B) General exposure to 488 nm laser to activate OptoDRONC in all *pnr*-expressing LECs promotes death. Time indicates hours after 488 nm laser exposure. Scale bar: 50 μm. Genotypes: (B) *ywhsFlp*/+; UAS-OptoDRONC::GFP/+; UAS-nls::mCherry, *pnr*-GAL4/+.
(PDF)

**S4 Fig. Analysis workflow of LEC clustered apoptosis.** (A) Workflow of cluster cell death detection and in silico simulation for randomized LEC apoptosis. Cell tracking data from TrackMate are set as the input for LEC neighbor detection via Delaunay Triangulation. Pairs dying concurrently are labeled, and if two of more pairs are interconnected, they are determined as cluster deaths. (B) Confocal imaging of nls::CFP::Venus of *shi*[TS] expressing LECs starting from 28 hAPF, 2 h before early phase at 30 hAPF. The first signs of clusters of cells preparing for apoptosis starts at 29 hAPF (white arrows) which completes with nuclear breakdown at 30 hAPF (yellow arrows). Scale bar: 100 μm. Genotypes: (B) *ywhsFlp*/+; *tubP*-miniCic::mScarlet/+; UAS-nls::CFP::Venus, *pnr*-GAL4/ UAS-*shi*[TS]. The data underlying the graphs shown in the figure can be found in https://zenodo.org/records/13290047.
(PDF)

**S5 Fig. Characterization of LEC clustered apoptosis.** (A) Representative confocal images and diagrams showing various shapes and sizes of clustered LEC apoptosis. Time stamp indicates time to apoptosis in hours. Scale bars 50 μm. (B) Boxplot of the number of cells in single LEC clustered apoptosis events between control, *EGFR*[RNAi], and *shi*[TS] pupae. Error bars are SEM. Mann–Whitney test, ***$P < 0.001$. (C) Confocal images of a cluster of apoptotic *EGFR*[RNAi] expressing LECs represented using cell outlines (neuroglian::GFP, NRG::GFP). Clustered apoptosis is outlined by cyan lines. White arrows indicate epithelial sealing upon the completion of clustered apoptosis. Yellow arrowheads indicate an expanding nearby histoblast nest. Time stamp indicates time to apoptosis in hours. Scale bar 50 μm. Genotypes: (A, B) *ywhsFlp*/+; *tubP*-miniCic::mScarlet /+; UAS-nls::CFP::Venus, *pnr*-GAL4/+. *ywhsFlp*/+; *tubP*-miniCic::mScarlet /+; UAS-nls::CFP::Venus, *pnr*-GAL4/ UAS-*EGFR*[RNAi]. *ywhsFlp*/+; *tubP*-miniCic::mScarlet /+; UAS-nls::CFP::Venus, *pnr*-GAL4/ UAS-*shi*[TS]. (C) *ywhsFlp*/+; *NRG::GFP*/+; *pnr*-GAL4/ UAS-*EGFR*[RNAi]. The data underlying the graphs shown in the figure can be found in https://zenodo.org/records/13290047.
(PDF)

**S6 Fig. Genetic knock-downs of the EGFR ligands did not affect LEC elimination significantly.** (A) nls::SCAT3 expression in, from top to bottom: control, *spi*[RNAi], *grk*[RNAi], and *Krn*[RNAi] pupae. Time indicates hours APF. Scale bars 50 μm. (B) Percentage of remaining LECs in control vs. from left to right: *spi*[RNAi], *grk*[RNAi], and *Krn*[RNAi] pupae. $n = 3$ pupae each. Error bars are SEM. Kolmogorov–Smirnov test vs. control. *P*-values are as indicated. (C) Swarmplot of LEC apoptosis counts at each time point for control vs. *spi*[RNAi] LECs. Each dot represents 1 cell. $n = 3–4$ pupae. Mann–Whitney test vs. Control. ***$P < 0.001$. Genotypes: (A–C) *ywhsFlp*/+; UAS-nls::CFP::Venus/+; *pnr*-GAL4/ +. *ywhsFlp*/+; UAS-nls::CFP::Venus/+; *pnr*-GAL4/ UAS-*spi*[RNAi]. *ywhsFlp*/+; UAS-nls::CFP::Venus/+; *pnr*-GAL4/ UAS-*grk*[RNAi]. *ywhsFlp*/+; UAS-nls::CFP::Venus/+; *pnr*-GAL4/ UAS-*Krn*[RNAi].
(PDF)

**S1 Movie. (Related to Fig 4).** Representative video of transient EGFR activity up-regulation during the early phase and its loss during the late phase control LECs. Yellow circles indicate apoptotic cells. White circles indicate neighboring cells in the early phase with transient EGFR up-regulation. Green channel is nls::CFP::Venus, Magenta is miniCic::mScarlet. Time stamps indicate hours after puparium formation. Scale bars: 25 μm.
(AVI)

**S2 Movie. (Related to Fig 4).** Representative video of the loss of transient EGFR activity up-regulation during the early phase in *EGFR* [RNAi] and *shi* [TS] expressing LECs and its loss during the late phase. Yellow circles indicate apoptotic cells. Yellow arrows mark the primary apoptotic cells for analyses. Green channel is nls:: CFP::Venus, Magenta is miniCic::mScarlet. Time stamps indicate hours after puparium formation. Scale bars: 25 μm.
(AVI)

**S3 Movie. (Related to Fig 5).** Transient EGFR up-regulation in survivor LECs adjacent to OptoDRONC-induced apoptotic LECs. Cyan ellipse indicates the 488 nm laser exposure. White circles indicate apoptotic cells. Note that several cells undergo apoptosis in response to the OptoDronc activation and the unmarked surviving LECs exhibit transient reduction of miniCic signal. Green channel is nls::mIRFP, Magenta is miniCic::mScarlet. Time stamp indicates the time to 488 nm exposure. Scale bar: 25 μm.
(AVI)

**S4 Movie. (Related to Fig 5).** Transient EGFR up-regulation is absent in LECs without any apoptotic events. Cyan ellipse indicates the 488 nm laser exposure. Green channel is nls:: mIRFP, Magenta is miniCic::mScarlet. Time stamp indicates the time to 488 nm exposure. Scale bar: 25 μm.
(AVI)

**S5 Movie. (Related to Fig 6).** Simulation movies of randomly removed points and clustering based on Delaunay Triangulation in Late Control, Early *EGFR* [RNAi], and Early *shi* [TS] pupae. Red pairs dots connected with red line are paired eliminations. Interconnected pairs are considered clustered eliminations.
(MP4)

**S6 Movie. (Related to Fig 7).** Comparison of LEC elimination rates between representative Control, *EGFR* [RNAi], and *vn* [RNAi] pupae. Scale bar: 25 μm.
(AVI)

## Acknowledgments

We thank Dr. Romain Levayer, the NIG-FLY Stock Center, the Bloomington *Drosophila* Stock Center, and the Vienna *Drosophila* Resource Center for providing the fly lines used in this study; and Dr. Tomohiko Taguchi for useful discussion.

## Author Contributions

**Conceptualization:** Kevin Yuswan, Xiaofei Sun, Daiki Umetsu.

**Data curation:** Kevin Yuswan, Xiaofei Sun.

**Formal analysis:** Kevin Yuswan, Xiaofei Sun, Daiki Umetsu.

**Funding acquisition:** Kevin Yuswan, Erina Kuranaga, Daiki Umetsu.

**Investigation:** Kevin Yuswan, Xiaofei Sun.

**Methodology:** Kevin Yuswan, Xiaofei Sun, Daiki Umetsu.

**Resources:** Kevin Yuswan, Erina Kuranaga, Daiki Umetsu.

**Supervision:** Erina Kuranaga, Daiki Umetsu.

**Visualization:** Kevin Yuswan.

**Writing – original draft:** Kevin Yuswan, Daiki Umetsu.

**Writing – review & editing:** Erina Kuranaga, Daiki Umetsu.

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
