## [Editor Report · Decision Letter 0]

7 Nov 2023

Dear Dr Umetsu, 

Thank you for submitting your manuscript entitled "Switching to clustered apoptosis induced by reduction of endocytosis and EGFR signaling accelerates Drosophila tissue remodeling" for consideration as a Research Article by PLOS Biology.

Your manuscript has now been evaluated by the PLOS Biology editorial staff as well as by an academic editor with relevant expertise and I am writing to let you know that we would like to send your submission out for external peer review.

Once your full submission is complete, your paper will undergo a series of checks in preparation for peer review. After your manuscript has passed the checks it will be sent out for review. To provide the metadata for your submission, please Login to Editorial Manager (https://www.editorialmanager.com/pbiology) within two working days, i.e. by Nov 09 2023 11:59PM.

Kind regards,

Ines

--

Ines Alvarez-Garcia, PhD

Senior Editor

PLOS Biology

---

## [Decision Letter · Decision Letter 1]

23 Dec 2023

Dear Dr Umetsu,

Thank you for your patience while your manuscript entitled "Switching to clustered apoptosis induced by reduction of endocytosis and EGFR signaling accelerates Drosophila tissue remodeling" was peer-reviewed at PLOS Biology. Please also accept my apologies for the delay in providing you with our decision. The manuscript has now been evaluated by the PLOS Biology editors, an Academic Editor with relevant expertise, and by three independent reviewers. 

As you will see, the reviewers find the conclusions novel and interesting, but they also raise several concerns that would need to be addressed before we can consider the manuscript for publication. They all have similar concerns, mainly regarding the formation of the clustered apoptosis and the link with the increase of cell elimination rate, which need to be further characterised. In addition, the reviewers ask for missing controls, statistics, acknowledging better the limitations of the study and several clarifications.

In light of the reviews, we would like to invite you to revise the work to thoroughly address the reviewers' reports. Given the extent of revision needed, we cannot make a decision about publication until we have seen the revised manuscript and your response to the reviewers' comments. Your revised manuscript is likely to be sent for further evaluation by all or a subset of the reviewers.

**IMPORTANT - SUBMITTING YOUR REVISION**

3. Resubmission Checklist

a) *PLOS Data Policy*

b) *Published Peer Review*

Sincerely,

Ines

--

Ines Alvarez-Garcia, PhD

Senior Editor

PLOS Biology

Reviewers' responses

Rev. 1:

In this study, Yuswan and colleagues characterised an interesting development switch which through the downregulation of endocytosis reduces EGFR/ERK activity, the occurrence of ERK pulses near dying cells and as such promotes the formation of clusters of cell elimination and accelerate the elimination of larval epidermal cells. This work is a follow-up from a previous study from the same authors which showed a global reduction of endocytosis levels during late pupal development, which was responsible for an acceleration of cell elimination. Moreover, this study recapitulates to a large extend a previous study performed in the pupal notum showing the existence of EGFR/ERK pulses near extruding/dying cells which inhibit cell death and prevent the formation of clusters of elimination. This work nicely complement these two previously published work by two main additions : 1) showing the role of endocytosis for the fluctuations of EGFR/ERK as well as the global downregulation of EGFR/ERK activity 2) showing for the first time the existence of a developmental switch that blocks the EGFR/ERK pulses near dying cells which as a consequence accelerate LECs elimination and allow the formation of clusters of cell elimination.

The study is well performed, with all the adequate control and quantifications (apart from few details for quantification and putative limitations, see below). It is also good to see that some of the process observed in the pupal notum can be recapitulated in the LECS to a great extent. Of note, the existence of EGFR/ERK pulses near extruding larval epidermal cell of the abdomen was already shown in Valon et al Dev Cell 2021 (Figure S4). It does not remove any novelty about the two main points I mention above, but I think it would be fair to mention it somewhere in the main text. I am overall very supportive for publication in Plos Biology as it opens very interesting perspective on physiological switch of apoptosis rate related to endocytosis and ERK dynamics, which may be relevant in other contexts/ tissues. Given the relative ubiquitous observation of ERK pulses and waves near dying cells (MDCK cells, MCF10A cells, HelA cells, and more recently human colon organoid), these results might be of very broad interest.

I still have though some important suggestions requiring text and figure editing, to match what can be really conclude firmly with the current data. I also have some suggestions to clarify some results and quantifications.

Main points :

1/ The authors made strong statements about the occurrence of clusters of cell elimination and their direct role for the acceleration of LECs elimination rate. While it is true that their data show a striking correlation between the two, it is actually pretty hard to distinguish what is the effect of a global increase of the rate of cell elimination (irrespective of its spatial distribution) and what is the contribution a specific spatial distribution of cell death. Unless that authors have a way to compare two conditions where global death rate is the same, but where the clustering is different, I would maybe suggest to clearly describe this limitation and tune down the statements in the title, abstract , results and discussion (and I don't think this will reduce the interest of this study). For instance, the short title "Clustered apoptosis regulates tissue remodeling rate" looks like a strong statement and I would rather use something more neutral like "A switch to cluster apoptosis is associated with accelerated tissue remodeling".

2/ Along the same line, this is currently difficult to distinguish what is the relative impact of ERK pulses near dying cells versus the global modulation of ERK basal activity. Both could equally contribute to an increase rate of cell elimination and as far as I can judge, the authors don't really have a condition that can sort these two parameters. I would therefore suggest to mention clearly this limitation in the discussion

3/ So far, the authors interpret the increase occurrence of clusters by the reduction of ERK pulses near dying cells, however if everything being random ( a pure Poisson process), the probability to observe cluster will also increase if the global rate of cell elimination is higher (independently of any local feedback), and could thus be explained by the change of the basal levels of ERK activity (see point 2). Is there any condition where the authors could see relatively mild effect on the global speed of elimination and yet very clear increase of the number of clusters ? That could help to argue for a specific effect on cell death distribution. Alternatively, one would need to go for a much more thorough description of the spatiotemporal distribution of cell death to prove that there is different "clustering" levels relative to a Poisson distribution with the same rate. Again, I don't think this is absolutely necessary here (and this was already performed in other context before) but I would as such remain cautious on this. Still, this could also be complemented by plotting the number of clusters in the different genetic background like in Figure 6I, but this time as a function of the global rate of cell elimination. That would help to partially clarify this.

Point 1,2,3 are admittedly hard to address in a very clean manner (unless one is lucky enough to find a conditions that help to disentangle these factors) and would require also quite thorough statistical characterisation of cell death distribution combined with modeling (to distinguish what is coming from a change of the absolute probability of cell death or what is driven by the direct impact of the local feedback). I don't think this change the overall interest of the study, but I think it would be fair to clearly acknowledge these limitations in the discussion and be cautious about statement along these points.

4/ Along the 3 points above, did the author look at the ERK pulses in the context of active Rab5 overexpression ? If ERK pulses are affected, while global ERK activity is higher, it would be an interesting case to discuss the point above.

Minor points:

1/ If I understood correctly, the optoDronc experiment was performed through a global activation of Dronc in the pannier domain. However, I guess it is difficult to sort in this case the cell autonomous effect of caspases from the non-cell autonomous effect (albeit I do agree with the argument of the authors about caspase activation being autonomously associated with ERK downregulation). Still, since the LEC cells are pretty large and pretty flat, it should be doable to activate Dronc in a single cell without using two photon excitation. Have the authors tried this type of experiment ? Admitedly, I don't think this result is really essential for their study and there is enough data showing a clear correlation between dying cells and ERK activation in the neighbours.

2/ Whenever using the miniCic sensor, the authors state (both in the main text, first time in line 162, and in graph axis of figures and legends) that this is a EGFR sensor. This is not quite correct since miniCic should be sensitive to ERK activity irrespective of the upstream activators. To be totally correct, I would suggest to change everywhere in the text/legends/figures this point and replace "EGFR levels" by "ERK activity".

3/ Unless I am mistaken, Figure S1 and Figure S4 are not referred to anywhere in the text.

4/ line 74: I am wondering if there is not a typo here. Shouldn't it be LEC number decrease (rather than increase) ?

5/ Line 217: are the authors talking about the increase of miniCic at late stage that can still be observed upon mirRHG treatment (the formulation was not perfectly clear for me) ? Also, do the authors have any idea/speculation on what could explain the global lower miniCic levels at late stage in mirRHG background ? (which cannot be explained by the absence pulses near dying cells which would go the other way around). That may suggest that low sublethal caspase activity may directly downregulate ERK.

6/ The measurement of ERK fluctuations are interesting, however the way it is done right now (Imax-Imin) should be greatly influenced by the absolute levels of miniCic signal (similar to any standard deviation that tends to scale with the mean). Would the authors obtain similar results it this would be normalised by the average levels of miniCic ? In a way, this would be similar to look at a coefficient of variation.

7/ For the optogenetic experiment, this is a detail, but I would maybe suggest another terminology in the text and figure ("bleach" "irradiated") that convey the wrong idea that one need very high laser power to obtain activation. Actually power similar to the one used for imaging should be sufficient to activate CRY2 (most of the optogenetic system can be activated at very low light intensity). I would suggest to use something more neutral like "blue light exposure".

8/ Figure 6I: there is a miss-match between the x axis title, the legend and the description in the main text (line 327). Are the value on x axis representing the average miniCic basal levels or are they representing the value of miniCic during pulses near dying cells ? I would suggest to clarify this point (both measurements would actually be relevant, again related to the main point 1, 2 and 3 above) .

9/ Line 440-441 : I would suggest to maybe reformulate the statement about artificial conditions. Indeed previous studies have characterised clustered elimination that appear upon perturbation of EGFR/ERK using exactly the same sort of genetic background and conditions used by the author here, so I was a bit surprised by this statement. However what is clearly new here is the "physiological" switch to clustered elimination that was never described before to my knowledge ( which is exactly what the authors point at in the next sentence). Unless the authors have a strong reason to keep this statement line 440, I would suggest to remove it (it is ambiguous and not really justified).

10/ Discussion : the authors describe an apparent contradiction between the requirement of a ligand and the putative role of mechanics in ERK activation. I would argue that the fact that the process is ligand dependent is not at all incompatible with a mechanical gated mechanism, specially since the authors have performed global Vein down-regulation (which might be globally permissive for ERK activation) and have not shown that this is specifically required in the dying cell. For instance, the ERK waves described during collective migration in MDCK cells are clearly mechanical dependent and yet also require several EGFR ligands (see work from Hirashima lab). I would suggest to be more nuanced on this statement and clearly mention the limitation of the experimental data regarding the role of Vein in dying cells versus a more global permissive function.

11/ Other studies have characterised ERK pulses near dying cells in the fly gut (Lucy O'Brien lab, Nature 2017, 10.1038/nature23678P), the human colon organoid (Pond et al. elife 2022, Paek lab, 10.7554/eLife.78837), or the secretion of FGF by dying HeLa cells (Bock et al. Nat Comm 2021, Tait lab, 10.7554/eLife.78837). I guess it would be fair to also quote them in this article..

Rev. 2:

Previously, the authors have shown that during abdominal morphogenesis, there are two phases of larval epithelial cell (LEC) elimination; an early phase, which is characterized by a slow elimination rate, and a late phase, which is characterized by a fast elimination rate (Hoshika et al., 2020). They also showed that this acceleration of cell elimination is due to a reduction in endocytic activity in the LECs, which regulates cellular myosin II organization, junctional E-cadherin levels, and caspase activation (Hoshika et al., 2020).

In this manuscript, Yuswan et al. now show that the acceleration of LEC removal is achieved by a change in apoptotic behavior of the LECs. They show that in the early phase, EGFR signaling is transiently upregulated in the neighbors of apoptotic cells, thus, limiting apoptosis to single-cell events. They also show that EGFR activity in the neighboring cells is EGF-ligand dependent. Lower endocytic activity and lower EGFR activity then promotes the switch to the late stage, in which LECs undergo clustered apoptosis where apoptosis is not limited anymore to single-cell events.

This manuscript establishes the role of EGFR signaling in the early and late stage well, however, the role of endocytosis in the process is less clear. The discovery of the switch between the two apoptotic modes is interesting. Some of the manuscript's results do not sufficiently support the conclusions, are not sufficiently interpreted, or are not sufficiently explained, as explained below:

Major comments

1. The paragraph 'EGFR signaling activity decreases over time in LECs in an endocytosis-dependent manner' needs improving:

- Line 166: '…low level, as shown by the nuclear localization of miniCic, which further increased over time' - This statement might not be sufficiently supported by the data. A statistical test needs to be done to show that indeed, in Fig. 2F, the observed increase in nuclear localization in the controls are indeed statistically significant.

- Line 167: 'This observation shows that EGFR decreases globally throughout LEC elimination consistent with the decrease in endocytic activity' - This manuscript does not show how endocytosis changes over time. Thus, Hoshika et al., 2020 should be cited here. Even better, an analysis of an endocytic marker from 20 to 35 hours APF could be presented (Hoshika et al., 2020 do not present data for the whole time period discussed here).

- Moreover, it could be made clearer that at this point in the manuscript, there is merely a correlation between the EGFR activity and endocytosis activity.

- Line 170: 'Abolishing endocytic activity by shiTS decreased average EGFR activity, especially in the early phase (Fig. 2D-F). - This statement needs to be improved, as only the early phase shows a significant difference compared to the control. Thus, the shits data does not sufficiently support the idea that EGFR activity and endocytosis activity correlate over time.

- Also, how I understand the presented data, the data can be interpreted so that shits leads to an early onset of LEC elimination, but that elimination itself is rather normal.

- Line 171: 'Consistently, we also observed an increase in average EGFR/ERK activity when endocytic activity was promoted by constitutively activating the endosome internalization regulating GTPase Rab5 (Rab5 Q88L) [36,37] (Fig. 2C, F)'. - This is imprecise, as there is no increase but the lack of a reduction in later stages. This suggests that there is a delay in elimination as stated by the authors in line 174: 'Rab5 Q88L significantly delayed LEC elimination compared to control for the time it took to reach half of the initial number of LECs(Fig. 2G-H).'.

- The sample size in Fig. 2h is too small for reliable statistics. At least 5-6 data points each are needed.

- Line 176: 'These results indicate that EGFR activity globally decreases over time and that the decrease of EGFR is regulated by the reduction of endocytic activity during LEC elimination.' - The data does not sufficiently support the conclusion that the decrease of EGFR activity is correlating with endocytic activity. The data can be interpreted in a way that EGFR activity and endocytic activity determine the onset of cell elimination.

- It would be important to include the results of the important controls shown in Fig. S1 in the text. This figure is not mentioned in the text.

2. The data of miniCic fluctuations in Fig. 3 are not sufficiently explained for the reader to interpret und to understand them. The methods section does not explain how the fluctuations were analyzed.

- How was fluorescence intensity normalized?

- What does Fig. 3A show?

- How was the degree of fluctuation calculated? Which time period was considered to calculate the data shown in Fig. 3B?

3. I am confused by the interpretation of the data shown in Fig. 3G. In line 216, the authors state that 'Interestingly, the average miniCic intensity increases up to 30h APF during LEC elimination (Fig. 3G), suggesting that the global EGFR/ERK activity decrease is independent of pro-apoptotic gene activity.', however, rhg-RNAi appears to affect miniCic nuclear localisation, suggesting that pro-apoptotic gene activity affects EGFR activity.

Also, the average miniCic intensity does not increase, as stated by the authors; it stays constant at a relatively high level without decreasing over time.

4. I do not understand the sentence in line 252: 'From these observations, we reasoned that fluctuations of EGFR/ERK may originate from the transient increase of EGFR/ERK in surrounding LECs.' - Do the authors mean that the observed fluctuations are mainly due to the fact that cells, that are neighbours of early cell death events, fluctuate?

5. This sentence is unclear: Line 257: 'Together, these results suggest that the fluctuations of EGFR/ERK activity is the reflection of its transient increase in cell clusters….'. Do the authors again mean that the observed fluctuations are mainly due to the fact that cells, that are neighbours of early cell death events, fluctuate?

6. A more detailed discussion of the results of Fig. 4F-l,J and how the authors interpret them is needed.

7. In the paragraph 'LECs neighboring a primary apoptotic LEC exhibit a transient increase in EGFR signaling activity', which begins in line 236, the authors show in Fig.4E that more cells die in the late phase, but this is not discussed in the text. It is only made clear that the behavior of the miniCic is changing in the neighbors of a dying cell, but not that the amount of dying cells changes. This is only explained later in the text. This is confusing.

8. In Figure 5, no n-numbers are given.

9. Why are the data for the controls at 18ºC and 25ºC so different in Fig. 6L?

10. The observation of cluster cell death is very interesting. It would be helpful to see data with a membrane marker to understand the impact of cluster cell death on the integrity of the epithelium. As far as I understand, neighboring LECs do not delaminate at the same time (probably to avoid a rupture in the tissue).

11. The cluster analysis is crucial for this manuscript. The way the clusters were identified means that, in a cluster of four cell deaths, two of the cell deaths might be two time points apart (and thus not clustered?). I wonder whether the way the analysis was done does merely reflect that, in the late stage, many more cells die and thus statistically more cells die in the vicinity of each other. Could the authors explain better how the analysis was done and how this provides evidence for clustering?

Also, some more information about the clusters would be helpful: Are there multipe clusters at the same time? Do clusters have a certain shape? How many cells form a cluster?

Moreover, the authors do not state the time interval in the movies used for the cluster analysis in Fig. 6. This is important to assess to what extent cell death in two consecutive frames can be considered to be dying at the same time.

Other comments

Line 139 'indicate that EGFR expression has an inhibitory effect against LEC elimination' should be 'indicate the EGFR activity has…', I think.

Fig. 1 - Graph in top right lacks a y-axis label. In E, wrong color for EGFRtop graph.

Line 166 '…low level, as shown by the nuclear localization of miniCic, which further increased over time' - It would be clearer if the authors would use two sentences to make this point clearer: '…low level, as shown by the nuclear localization of miniCic (Fig. 2B, F)). This nuclear localization further increased over time (Fig. 2B, F)'.

Fig. 2F - The y-axis label could be clearer, e.g. 'Ratio nuclear/cytoplasmic miniCic intensity'.

Line 199: '…fluctuates differently for each cell (Fig. 3A top, B top).' - should only be 'B'.

Is Fig. 3D showing nls::CFP::Venus (stated in the figure) or nls::SCAT3 as stated in the legend?

Rev. 3:

In this manuscript, Yuswan and the team investigate the apoptosis mechanism of larval epidermal cells (LECs) during Drosophila metamorphosis. Expanding on their prior discovery of decreased endocytic activity in LECs leading to increased apoptosis, the authors show the crucial role of the epidermal growth factor receptor (EGFR) signaling pathway in governing both endocytic activity and LEC apoptosis. Additionally, they demonstrate that EGFR in neighboring cells prevents cluster apoptosis. Although this protection mechanisms via ERFR signaling were documented in other contexts, this study introduces a novel concept where developing tissue regulates apoptosis modes to meet specific developmental needs, such as the requirement for clustered apoptosis to facilitate the growth of neighboring tissue. Notably, this is distinct from the traditional view that cluster of apoptosis is disadvantage for maintaining tissue barrier function.

While the research question is intriguing, concerns arise about the clarity of explanations and supporting evidence for critical data, suggesting that the manuscript may be somewhat premature in its current stage.

MAJOR COMMENTS:

#1. The use of two different methods to analyze EGFR activity introduces confusion. Analyzing some data as the ratio of nuclear to miniCic intensity (Figs. 2, 3, 6) while analyzing others as miniCic intensity (Figs. 4, 5) makes it challenging to intuitively link the two measures. For example, when miniCic intensity increases, the ratio indicating ERK activity could either rise or fall, depending on alterations in nuclear intensity. Conversely, the value of miniCic intensity, representing ERK activity, increases as the miniCic intensity of the image rises. To enhance clarity, I recommend maintaining consistency in the analyses, unless there are specific reasons for the variation, which should be explicitly clarified in the text. A side note: Fig. S1 uses nuclear to 'cytoplasmic' miniCic intensity as the third measure of EGFR activity.

#2. In Fig. 2B, it is hard to see the distribution of miniCic as shown in Fig. 2A. To enhance visibility, it would be beneficial to include a composite image of the green (nls::CFP::Venus) and purple (miniCic::mScarlet) channels, at least for the wild-type. This additional representation would provide a clearer visual understanding of the miniCic localization, especially in the cytoplasm.

#3. It is challenging to remember high nuclear/miniCic intensity represent low EGFR activity and vice versa. It would be helpful to include an indicator, similar to the one in Fig. 4E, in Fig. 1F.

#4. Fig. 3A. It raises questions about the observation of more blue (indicating high EGFR activities) during 30h and 35h APF. This is opposite from the trend shown in Fig. 2F. Please clarify.

#5. Fig. S2D. Including a plot of miniCic intensity for surviving cells would be beneficial. This additional data would provide insights into the timing of EGFR activities in surviving cells relative to caspase activation in dead cells.

#6. In Fig. 4A, EGFR/ERK upregulation is observed in several neighboring larval epidermal cells (LECs); however, not all neighboring cells exhibit this upregulation. It would be insightful to discuss potential mechanisms for this differential response in the discussion section, addressing why some neighboring cells show upregulation while others do not.

#7. It would be helpful to show that tissue integrity remains uncompromised following cluster apoptosis by including E-cad data fomr the wild-type late phase.

#8. It's not clear why most of the data starts from 20h APF, but Fig. 1I begins from 30h APF? To ensure consistency, it would be helpful to either explain the reasoning for this discrepancy or provide data from 20 hours APF for Fig. 1I.

#9. While I understand that optoDRONC induces apoptosis in several cells but not single cell, the manuscript lacks information regarding the extent of the affected area through optogenetics. It would be valuable to estimate the size of the region where optoDRONC can induce apoptosis. Consider expressing optoDRONC in other tissues, such as histoblasts, and employing the same laser settings to monitor the number of cells undergoing apoptosis. This approach could provide insights into the spatial reach of optoDRONC in inducing apoptosis.

#10. It's hard to see "transient EGFR/ERK activation (line 287)" in the figure. To enhance clarity, I recommend making this phenomenon more visually by incorporating arrows, ellipses, or other graphical elements.

#11. The statement that "EGFR/ERK activity increases locally and transiently in the cells adjacent to an apoptotic cell (line 390)" is made, but to support this claim, I recommend checking whether the next-nearest cells do not exhibit an increase in EGFR/ERK activity.

#12. It is not clear which data shows "vn regulates the transient EGFR activation in cells adjacent to the apoptotic LEC (line 363)". The reduction in EGFR fluctuation with vn RNAi, as shown in Fig. 7E, is not specific to the cells next to the apoptotic cell. Additionally, the statement, "We further identified that the EGFR ligand vein (vn) is a key modulator of the transient EGFR/ERK upregulation that promotes the survival of cells neighboring apoptotic LECs (lines 398-400)," appears to be an overstatement. To ensure accuracy, it is recommended to rephrase the text.

#13. In the discussion, particularly from line 407 to 432, there is a mixing of perspectives, shifting from a global standpoint (ecdysone) to a local one (caspase) and then returning to a global perspective (global change in junctional tension). This creates confusion for the reader, making it challenging to understand the connection between the discussion points and either global ERK decrease or local ERK fluctuation. To enhance clarity and coherence, I recommend rearranging the order of this section of the discussion, ensuring a logical flow that helps the reader better follow the linkages between the different perspectives on ERK regulation.

MINOR COMMENTS:

#14. The labels on the x- and y-axes are either missing or unclear in the following figure.

- Fig. 1A right

- Fig. S2D. What's the unit?

- Fig. 6F-H. What's the unit?

#15. Some still images and graphs use hours as the unit, while others use minutes. Please standardize the unit throughout for consistency.

#16. Fig. 2C-E. Change the order so that it aligns with the sequence in which they appear in the text. In the text, Rab5^ Q88L appears later than shi^TS.

#17. It would be informative to include heatmaps depicting the nuclear miniCic::mscarlet intensity over time for rhg^RNAi, similar to the presentation in Fig. 3A.

#18. The following sentence lacks statistical support.

- "the average miniCic intensity increases up to 30h APF… (line 216)"

#19. "Previous reports have observed the upregulation of EGFR/ERK in clusters of cells in response to apoptotic events [30,38], and our data also indicate that the fluctuation of EGFR/ERK requires the initiation of LEC apoptotic events (line 238)." The mention of EGFR in both neighboring cells and apoptotic cells in a sentence can be confusing.

#20. "We observed a transient increase of EGFR/ERK in … (line 248)." Adding the clarification about the transient decrease of miniCic intensity, such as 'a transient increase of EGFR/ERK, accompanied by a transient decrease in miniCic intensity, in …,' enhances the reader's understanding.

#21. It appears that Fig. 4A (and 4B) may not correspond to Movie 1 (and 2). I recommend using the same data and magnification for both the still images in the figure and the movies to ensure consistency.

#22. It would be helpful to indicate the primary apoptotic cell in Movie 1 and 2.

#23. I would suggest revising the following wordings.

- "within roughly 60 minutes of primary LEC apoptosis (line 249)". The word 'within' is confusing. It would be more clear to use 'proceeding'.

- "EGFR upregulation in the LECs that initiated apoptosis (line 289)". It would be clearer to state 'EGFR upregulation in the LECs associated with induced apoptosis'.

- "we reasoned that fluctuations of EGFR/ERK may originate from the transient increase of EGFR/ERK in surrounding LECs (line 252)". Please specify which fluctuation are you refiring to about the first 'fluctuation' in this sentence. Is it fluctuation in the primary dying cell?

- "non-linearly changes the state of cells." It is not clear what 'non-linearly' means.

#24. n number is missing in Fig. 5.

#25. I don't see the term 'cyan ellipses' mentioned in the legend of Fig. 5 corresponding to the images.

#26. What does 'the degree of transient EGFR activity... (Fig. 6I)' mean, and how does it relate to the content presented in Fig. 6I?

#27. I suggest overlapping the data of EGFR RNAi with Fig. 7F to maintain consistency with the other graphs in Fig. 7. The same comment applies to Fig. 6E.

#28. Only scientists involved in histoblast expansion may comprehend 'the proliferating cells of neighboring tissue, which increase exponentially and thereby replace the shrinking tissue more rapidly as tissue remodeling progresses (line 444).' I recommend adding more detailed information, including 'Histoblast', to facilitate understanding for non-experts.

#29. I don't see a "sharp transition (line 449)" in the phenomenon observed.

#30. Typo

- Reference (lines 92 and 100).

- Fig. 2G (line 137) should be Fig. 1G.

- Fig. 2H (line 138) should be Fig. 1H.

- Fig. 3D-E (line 211) should be Fig. 3C-E.

- Fig. 6C (line 282) should be Fig. 5C.

#31. The final sentence of the abstract, "These findings reveal the tight temporal regulation of tissue remodeling, accomplished by mode switching via local cell-cell communication." is too general and doesn't adequately signify the novelty of the study. I suggest revising this sentence to better capture the unique contributions of the research. Please see my summary paragraph above for more detail.

---

## [Decision Letter · Decision Letter 2]

28 May 2024

Dear Dr Umetsu,

Thank you for your patience while we considered your revised manuscript entitled "Switching to clustered apoptosis induced by reduction of endocytosis and EGFR signaling accelerates Drosophila tissue remodeling" for consideration as a Research Article at PLOS Biology. Your revised study has now been evaluated by the PLOS Biology editors, the Academic Editor and the three original reviewers. 

The reviews are attached below. You will see that the reviewers appreciate the revisions done in the manuscript, however they still raise several issues that would need to be addressed in a final revision. Reviewer 1 is not convinced about the physiological function of the clustering of cell death, but acknowledges it is probably difficult to confirm, so you could alternatively tone down the title. This reviewer also requests additional simulations to show the relative contribution of basal ERK activity vs ERK pulses for the regulation of cell death cluster in the late pupae, EGFR RNAi and ShiTS conditions. Reviewer 2 thinks you should provide a better characterisation of the phenomenon and integrate in the manuscript the points discussed in the rebuttal. In addition, this reviewer suggests you should repeat some of the simulations considering that clustering is not due to an increase in cell death rate, and also cell outline data showing how clustered apoptosis impacts cell removal and tissue integrity. Reviewer 3 suggests using TrackMate to measure the nuclear miniCic intensity for some of the data.

In light of the reviews, we are pleased to offer you the opportunity to address the remaining points from the reviewers in a revision that we anticipate should not take you very long. We will then assess your revised manuscript and your response to the reviewers' comments with our Academic Editor aiming to avoid further rounds of peer-review, although might need to consult with the reviewers, depending on the nature of the revisions.

We expect to receive your revised manuscript within 1 month, but please email us (plosbiology@plos.org) if you have any questions or concerns, or would like to request an extension if you need more time.

**IMPORTANT - SUBMITTING YOUR REVISION**

3. Resubmission Checklist

a) *PLOS Data Policy*

* Supplementary files (e.g., excel). Please ensure that all data files are uploaded as 'Supporting Information' and are invariably referred to (in the manuscript, figure legends, and the Description field when uploading your files) using the following format verbatim: S1 Data, S2 Data, etc. Multiple panels of a single or even several figures can be included as multiple sheets in one excel file that is saved using exactly the following convention: S1_Data.xlsx (using an underscore).

* Deposition in a publicly available repository. Please also provide the accession code or a reviewer link so that we may view your data before publication.

Fig. 1A, E, I; Fig. 2F-H; Fig. 3A, B, E-H; Fig. 4F-J; Fig. 5D, E; Fig. 6E, F, G; Fig. 7C-G; Fig. S1D; Fig. S2D; Fig. S4C and Fig. S5B, C

***Please also note that for data deposited in Github, you should obtain a doi in Zenodo. You can follow this instructions: 

https://cassgvp.github.io/github-for-collaborative-documentation/docs/tut/6-Zenodo-integration.html

b) *Published Peer Review*

Sincerely,

Ines

--

Ines Alvarez-Garcia, PhD

Senior Editor

PLOS Biology

Reviewers' comments

Rev. 1:

I appreciate the effort of the authors to answer my points and they have now included interesting complement in the current manuscript which is really improved. I should say though that as a matter of fact, my points 1 and 3 that I raised are not fully sorted by what is proposed by the authors. I am sorry to insist on those, but since most of the articles is about the clustering and ERK dynamics, I believe they need to be well set.

In point 1, I was arguing that it is difficult to prove that the clustering of cell death has a physiological function (and accelerates tissue remodeling) as claimed by the authors. To prove this, one would need to compare two conditions with exactly the same death rate, but with one randomly distributed, and one clusterised, and see how this affect the developmental process. Without this, all we can say is that there is a correlation between the appearance of clusters and an acceleration of LEC elimination, and this is not possible to state any causal link. So I remained convinced that it is very hard to prove that the "clustering" of cell elimination has a function per se. This does not remove the interest of most of the observations, but I still believe that the title of the manuscript is misleading as there is no proof that cell death clustering "accelerates" tissue remodeling (the best one can say is that clustered death "is associated with" accelerated tissue remodeling).

In point 3, I was pointing at the difficulty to sort the contribution of global ERK modulation and ERK pulses and their link with clustered death. The authors provide now interesting simulations showing that clustered death is much more frequent than what would be expected from a random distribution, and this for any condition (even for the control at early stage). This is per se very interesting as it suggests that there is a natural tendency to have clustered apoptosis (maybe because of underlying pattern that kill cells in specific domains ?), and open another unresolved question of why is there higher clustering than expected from random. However, this does not really address my point which is to sort the relative contribution of basal ERK activity (and thus basal rate of cell death) versus ERK pulses for the regulation of cell death cluster in the late pupae, EGFR depleted and shiTS conditions. To really address this point, the authors could maybe compare the relative proportions of clusters in between control, EGFR RNAi and ShiTS in the simulations (which will estimate the fold increase of clusters number expected just based on changes in the rate of cell death), and then do the same for experimental data from Figure 6E (looking at fold increase of clusters relative to the WT). Assuming that ERK fluctuations plays some role in cluster occurrence, then one would expect higher fold change of cluster numbers in EGFR RNAi and ShiTS relative to experimental control compared to the fold change obtained in the simulations (in summary, comparing "relative" frequency compared to control and not absolute numbers, which hide difference because of this natural tendency of clustered death). Right now, the data in Figure 6G would tend to suggest that most of the clustering behaviour could be explained by ERK basal activity (and thus is not really giving strong argument for the contribution of ERK fluctuations/pulses in the regulation of death clustering).

Other minor points :

- The text line 86 was a bit missleading : "there is a global EGFR/ERK decrease throughout the LEC elimination period. The dynamics of the decrease in EGFR/ERK parallels with the decrease in endocytic activity in LECs, which was reported in our previous work [14]. While the reduction of endocytic activity ceases around 26 hAPF during the transition into the late phase [14], we observed that EGFR/ERK activity continuously decreases even during the late phase."

It sounds like the authors already showed that there was a reduction of ERK in the LEC cells in ref 14, but this is what is shown here in this paper. Was this piece of text intended to come later in the intro (when summarising the results of this study)?

Rev. 2:

I very much appreciate the authors' responses to my comments. The authors have addressed many of my comments. However, the authors have not sufficiently addressed my concerns about the characterisation of the clustered cell death and its analysis.

1) In my opinion, a better characterisation of the phenomenon is needed for the reader to fully understand it. (A few of the points have been addressed by the authors in their response to the reviewers, but they should also be presented in the main text of the manuscript.)

a) The position of cell death (as well as of clusters) in the tissue needs to be characterised in more detail. The authors have the tracking data to create maps with the position of cell death in 2D over time. This will help to assess where cell deaths and clusters appear. Do they appear anywhere in the tissue, closer to the histoblasts, or closer to the midline? This is important, as cell death is not randomly distributed in the tissue - more LECs die close to the histoblasts than further away (Nakajima et al., 2011).

b) In Fig. 6E, the authors only show the number of cell deaths but not the percentage of total death. This is shown in Fig. S4C, but it might be better to show this in the main figure as well.

c) Are there multiple clusters at the same time?

d) Do the authors ever classify a cell as belonging to two clusters? This would impact the precision of their analysis.

e) A more detailed description of the various shapes of the clusters would be helpful (currently, there is only Fig. S4A). Also, are there certain shapes that are more common?

2) The authors have not fully addressed my comment on their choice of a 10-minute interval between frames. I was querying whether it is actually okay to argue that cells die at the same time (as a cluster), when indeed there might be a 10-minute difference between their deaths.

Along the same lines, is nuclear breakdown a good marker to assess timing of the cells' death? How fast are the changes in fluorescence and how sure can the authors be that cells with a similar fluorescence are indeed at the same stage of 'their death'? One way to test this could be to do the cluster analysis with a caspase reporter.

3) I am not sure whether the authors' simulation of a random distribution of cell death is sufficient to establish that clustering is not due to a mere increase in cell death rate. The reason is that cell death is not randomly distributed in the tissue (there are more cell deaths closer to the histoblasts (Nakajima et al., 2011)). What happens when the authors consider this in their simulations?

Also, as the authors analyse two segments, did they ensure that, when simulating cell removal, they treated the two segments independently? Otherwise, they would have a scenario where they removed one cell from one segment and one from the other, which would never result in neighbouring cells dying. This would skew the analysis. Moreover, it might be good to state the cell death rates used for the simulation in the methods.

4) Although I understand the authors' concerns about including cell outline data in this manuscript, I think that the description of cluster apoptosis is linked with the issue of tissue integrity and that this needs to be addressed in this manuscript. Thus, it would be important to include cell outline data showing how clustered apoptosis impacts on cell removal and tissue integrity.

In addition, looking at cell outlines will help to establish neighbourhoods and improve the characterisation of the phenomenon (see comment 1).

The analysis should be more detailed than what the authors have provided in the figure in the response to the reviewers and include an assessment of how many neighbouring cells delaminate at (almost) the same time and how this relates to clustered apoptosis.

5) When discussing ERK activity, the authors consider one cell with its six direct neighbours (scheme in Fig. 4E). How can this lead to the formation of clusters in the shape of a line of four cells?

Other comments:

1) I am very sorry that I did not spot this in my first review, but I have noticed that, using pnr-Gal4, it appears that not all LECs are labelled. How do the unlabelled cells behave? At least, it should be stated in the text that not all LECs were considered.

2) I would like to thank the authors for clarifying the point about the data from 18ºC and 25ºC (Fig. 6L). I think that the temperature dependence should be mentioned in the main text.

Rev. 3:

The authors have adequately responded to most of my comments. I have only a minor suggestion.

Regarding their response to comment #1:

I understand the challenges of tracking the ratio of nuclear to cytoplasmic miniCic intensity. Alternatively, I would suggest using TrackMate to measure the nuclear miniCic intensity for a few data in Figures 2, 3, and 6, for instance Fig. 2F control, and qualitatively cross-check whether the tendency observed from the ratio is consistent with the one using the nuclear intensity. Plotting such data side by side could be a helpful supplemental data.

Otherwise, I congratulate the authors on this intriguing study.

---

## [Editor Report · Decision Letter 3]

31 Jul 2024

Dear Dr Umetsu,

Thank you for your patience while we considered your revised manuscript entitled "Switching to clustered apoptosis induced by reduction of endocytosis and EGFR signaling is associated with the acceleration of tissue remodeling rate" for publication as a Research Article at PLOS Biology. This revised version of your manuscript has been evaluated by the PLOS Biology editors and the Academic Editor.

Based on our Academic Editor's assessment of your revision, we are likely to accept this manuscript for publication, provided you satisfactorily address the data and other policy-related requests stated below.

In addition, we would like you to consider a suggestion to improve the title:

"Reduction of endocytosis and EGFR signaling promotes a switch from isolated to clustered apoptosis during epithelial tissue remodeling in Drosophila”

We expect to receive your revised manuscript within two weeks. 

*Published Peer Review History*

*Press*

Sincerely,

Ines

--

Ines Alvarez-Garcia, PhD

Senior Editor

PLOS Biology

Fig. 1A, E, I; Fig. 2F-H; Fig. 3A, B, E-H; Fig. 4F-J; Fig. 5D, E; Fig. 6E, F, G; Fig. 7C-G; Fig. S1D; Fig. S2D; Fig. S4C and Fig. S5B, C

CODE POLICY

Please ensure that the code is sufficiently well documented and reusable, and that your Data Statement in the Editorial Manager submission system accurately describes where your code can be found.

---

## [Editor Report · Decision Letter 4]

30 Aug 2024

Dear Dr Umetsu,

Thank you for the submission of your revised Research Article entitled "Reduction of endocytosis and EGFR signaling is associated with the switch from isolated to clustered apoptosis during epithelial tissue remodeling in Drosophila" for publication in PLOS Biology. On behalf of my colleagues and the Academic Editor, Nic Tapon, I am delighted to let you know that we can in principle accept your manuscript for publication, provided you address any remaining formatting and reporting issues. These will be detailed in an email you should receive within 2-3 business days from our colleagues in the journal operations team; no action is required from you until then. Please note that we will not be able to formally accept your manuscript and schedule it for publication until you have completed any requested changes.

PRESS

Sincerely, 

Ines

--

Ines Alvarez-Garcia, PhD

Senior Editor

PLOS Biology
